# An orphan protein of *Fusarium graminearum* modulates host immunity by mediating proteasomal degradation of TaSnRK1α

Cong Jiang [1,2], Ruonan Hei[1], Yang Yang[1], Shijie Zhang[3], Qinhu Wang[1], Wei Wang[1], Qiang Zhang[1], Min Yan[1], Gengrui Zhu[1], Panpan Huang[1], Huiquan Liu [1✉] & Jin-Rong Xu [2✉]

*Fusarium graminearum* is a causal agent of Fusarium head blight (FHB) and a deoxynivalenol (DON) producer. In this study, *OSP24* is identified as an important virulence factor in systematic characterization of the 50 orphan secreted protein (*OSP*) genes of *F. graminearum*. Although dispensable for growth and initial penetration, *OSP24* is important for infectious growth in wheat rachis tissues. *OSP24* is specifically expressed during pathogenesis and its transient expression suppresses BAX- or INF1-induced cell death. Osp24 is translocated into plant cells and two of its 8 cysteine-residues are required for its function. Wheat SNF1-related kinase TaSnRK1α is identified as an Osp24-interacting protein and shows to be important for FHB resistance in TaSnRK1α-overexpressing or silencing transgenic plants. Osp24 accelerates the degradation of TaSnRK1α by facilitating its association with the ubiquitin-26S proteasome. Interestingly, TaSnRK1α also interacts with TaFROG, an orphan wheat protein induced by DON. TaFROG competes against Osp24 for binding with the same region of TaSnRKα and protects it from degradation. Overexpression of TaFROG stabilizes TaSnRK1α and increases FHB resistance. Taken together, Osp24 functions as a cytoplasmic effector by competing against TaFROG for binding with TaSnRK1α, demonstrating the counteracting roles of orphan proteins of both host and fungal pathogens during their interactions.

[1] State Key Laboratory of Crop Stress Biology for Arid Areas and NWAFU-Purdue Joint Research Center, College of Plant Protection, Northwest A&F University, 712100 Yangling, Shaanxi, China. [2] Department of Botany and Plant Pathology, Purdue University, West Lafayette, IN 47907, USA. [3] School of Life Sciences, Zhengzhou University, 450001 Zhengzhou, Henan, China. ✉email: liuhuiquan@nwsuaf.edu.cn; jinrong@purdue.edu

Comparative genome studies have revealed that fungal pathogens have a cadre of orphan genes that are restricted to a single species or narrow clade. Although the vast majority of them are of unknown functions, these taxonomically restricted orphan genes are thought to play important roles in lineage-specific adaptations[1–3]. Plant pathogenic fungi may evolve novel orphan genes to facilitate their infection or enhance virulence. Fungal effectors are good examples of orphan genes that have evolved for plant infection as many of them lack homologs in closely related species. To date, effectors of various molecular mechanisms have been identified and characterized in different fungi[4,5]. Most of them are small secreted proteins that are cysteine-rich but lack a common structural motif. In *Magnaporthe oryzae*, a model fungal pathogen, whereas apoplastic effectors are usually secreted via the conventional endoplasmic reticulum to Golgi route and accumulated in the apoplast, some cytoplasmic effectors are able to be secreted through the biotrophic interfacial complex (BIC) and translocated into plant cells[4,6]. The targets of fungal effectors also vary significantly in their functions, including transcription factors, protein kinases, and proteins or compounds that are involved in plant defense, signaling, and metabolic pathways[7–10]. For example, the Tin2 and Cmu1 effectors of *Ustilago maydis* target the ZmTTK1 kinase for affecting cell wall lignification and chorismate for affecting the metabolic status of infected plant cell, respectively[11–13].

*Fusarium graminearum* is a causal agent of Fusarium head blight (FHB), which is one of the most important diseases of wheat and barley worldwide. It develops compound appressoria and infection cushions for plant penetration[14]. After penetration, invasive hyphae grow inter-cellularity and intra-cellularly in infected plant tissues and develop bulbous, irregular invasive hyphae that are morphologically distinct from epiphytic hyphae[15,16]. Infectious growth then spreads from the initial infection site to neighboring spikelets on the same wheat heads via the rachis, resulting in the blight of entire wheat heads. In the past decade, many genes important for different infection processes have been identified in *F. graminearum*[17–20], including the putative *FGL1* and *FgNahG* effector genes. *FGL1* encodes a secreted lipase that can release free fatty acids to inhibit innate immunity-related callose formation during wheat head infection[21]. *FgNahG* is predicted to encode a secreted salicylate hydroxylase that can covert salicylic acid (SA) to catechol. The *FgnahG* deletion mutant was reduced in virulence and expression of *FgNahG* in Arabidopsis reduced its resistance against *F. graminearum*[22]. Recently, a non-ribosomal octapeptide, fusaoctaxin A, was identified as a virulence factor that is required by *F. graminearum* for cell-to-cell invasion in wheat coleoptiles[23].

Besides causing severe yield losses, *F. graminearum* is a producer of the trichothecene mycotoxin deoxynivalenol (DON)[24,25]. As an inhibitor of eukaryotic protein synthesis, DON is also phytotoxic and an important virulence factor. The *TRI5* gene, which is essential for DON biosynthesis, is expressed in infection cushions during early stages of infection[14]. The *tri5* deletion mutant is defective in spreading from the infected florets to other spikelets on the same head via the rachis[26] and transgenic plants expressing a UDP-glucotransferase gene that glycosylates DON are increased in resistance against FHB[27]. Overexpression of TaFROG that encodes a *Pooideae*-specific orphan protein induced by DON treatment also increased resistance against *F. graminearum*[28]. In wheat, TaFROG interacts with the TaSnRK1α protein kinase[28] and a NAC transcription factor[29]. Although germplasm with complete resistance or immunity to *F. graminearum* is lacking, more than 50 quantitative trait loci (QTLs) conferring various degrees of FHB resistance have been identified in wheat. To date, only *Fhb1*, a QTL that confers resistance to pathogen spread but not the initial infection, has been

characterized at the molecular level. The candidate *Fhb1* genes that have been reported include *TaPFT* encoding a pore-forming toxin-like protein[30] and *TaHRC-R* or *His^R* encoding a histidine-rich calcium-binding protein[31,32]. However, the underlying mechanism of these candidate *Fhb1* genes for FHB resistance remains to be characterized.

The *F. graminearum* genome is predicted to encode hundreds of orphan genes[33] but their roles in the pathogenic interaction and co-evolution of *F. graminearum* with its host plants remain to be characterized. Because small secretory proteins may function as effectors to interfere with plant immunity during infection[34–36] and no cytoplasmic effectors have been identified in *F. graminearum*, in this study we systematically characterized all the 50 genes encoding orphan secretory proteins (OSPs) and identified *OSP24* as an effector that is specifically expressed during plant infection and important for infectious growth in the rachis tissues of infected wheat heads. Transient expression of Osp24 in plant cells suppressed BAX-induced or INF1-induced program cell death. Furthermore, we identified the SNF1-related kinase TaSnRK1α as an Osp24-interacting protein and showed that TaSnRK1α is important for resistance against *F. graminearum* by RNA silencing or over-expression in transgenic plants. Osp24 may promote the degradation of TaSnRK1α through the ubiquitin-26S proteasome system[37]. TaFROG competed against Osp24 for binding to the same region of TaSnRK1α. Overexpression of TaFROG in wheat stabilized TaSnRK1α and increased resistance against *F. graminearum*. Overall, results from this study showed that the OSP Osp24 in *F. graminearum* functions as a cytoplasmic effector targeting TaSnRK1α for degradation but the wheat orphan protein TaFROG competes with Osp24 for binding with TaSnRK1α and prevents its degradation, indicating the evolving and active adoption of orphan proteins in the arms race between the pathogen and its host.

## Results

**Identification and characterization of OSP genes in *F. graminearum*.** Among all the predicted protein-coding genes in the *F. graminearum* genome, 971 of them (~7.3%) were identified as orphan genes using the bioinformatics analyses described in the "Methods" section. On average, these orphan genes encode smaller proteins than genes conserved in other species (Fig. 1a) and tend to be less transcribed or transcribed at lower levels (Fig. 1b) based on published RNA-seq data[20,30]. Fifty of them were predicted to encode proteins with signal peptides (SPs) and named OSP genes in this study. For a number of these OSPs, the mature peptides have fewer than 100 amino acid (aa) residues and are cysteine-rich (Fig. 1c), which is similar to the general characteristics of fungal effector proteins[4,5].

To determine their functions, gene replacement mutants were generated for the 50 *OSP* genes by the split-marker approach in the wild-type strain PH-1 (Supplementary Table 1, Supplementary Fig. 1). All the resulting mutants were normal in vegetative growth, conidiation, and sexual reproduction (Fig. 1d; Supplementary Table 2). In wheat head infection assays, most of the mutants were normal in virulence (Supplementary Table 2). However, mutants deleted of three individual *OSP* genes, *OSP24* (FGSG_11564), *OSP25* (FGSG_11647), and *OSP44* (FGSG_13464), were significantly reduced in virulence in repeated infection assays with wheat heads of cultivar Xiaoyan 22 (Fig. 1e). On average, the disease index of the *osp24*, *osp25*, and *osp44* deletion mutants was reduced 51%, 37%, and 36%, respectively, in comparison with that of PH-1 (Fig. 1f). The *osp24*, *osp25*, and *osp44* mutants also were reduced in virulence in infection assays with wheat cultivar Zhoumai 36[38] (Supplementary Fig. 2), indicating that the role of *OSP24* in plant infection is not cultivar-specific. When assayed with the inoculated

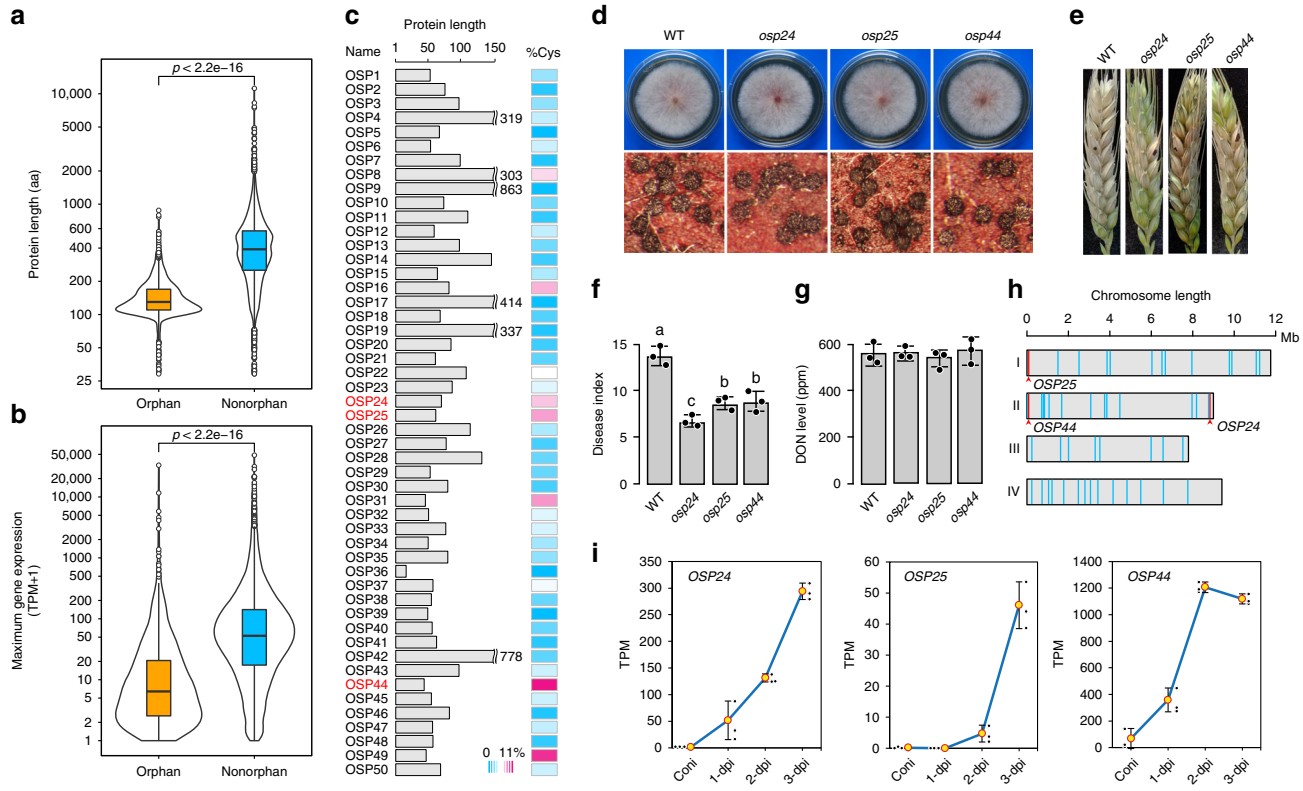

**Fig. 1 Characterization of orphan genes encoding secreted proteins in *Fusarium graminearum*. a** Comparative analysis of the length of proteins encoded by orphan and non-orphan genes with the Wilcoxon rank-sum test ($p < 2.2e{-}16$). **b** Comparative analysis of the expression levels of orphan and non-orphan genes with the Wilcoxon rank-sum test ($p < 2.2e{-}16$). For each gene, the maximum gene expression level was estimated from its transcripts per kilobase million (TPM) in RNA-seq data of conidia, vegetative hyphae, perithecia, and infected wheat heads. **c** Length and Cys (cysteine) content (%) of mature orphan secretory proteins (OSPs). The three OSPs important for virulence are in red. **d** Representative images of 3-day-old PDA cultures and 8-dpf (8 days post-fertilization) mating plates of the wild-type strain PH-1 (WT) and the *osp24, osp25*, and *osp44* deletion mutants. **e** Representative images of wheat heads infected with marked strains were photographed at 14 dpi. **f** The disease index of WT and three *osp* mutants. Error bar represents standard deviation (SD) from mean (marked with black dots on the bars) of three independent experiments ($n = 3$) with at least 10 wheat heads examined in each experiment. Different letters indicate significant differences based on ANOVA analysis followed by Duncan's multiple range test ($P = 0.05$). **g** Mean and standard deviations of DON levels in diseased wheat spikelets inoculated with WT or three *osp* mutants based on data from three biological replicates ($n = 3$). No significant differences was observed based on ANOVA analysis followed by Duncan's multiple range test ($P = 0.05$). **h** Distribution of the *OSP* genes (blue vertical bars) on the four chromosomes of *F. graminearum*. Red bars and arrowheads indicate the positions of *OSP24, OSP25*, and *OSP44*. **i** Expression levels of the indicated *OSP* genes based on their TPMs in RNA-seq data of conidia and infected wheat heads[20] sampled at 1-dpi, 2-dpi, or 3-dpi. Error bar represents SD from three biological replicates ($n = 3$).

spikelets sampled at 14 dpi, DON production was normal in these three mutants (Fig. 1g). The *osp24* mutant that had the lowest virulence was selected to assay for *TRI5* gene expression. In the inoculated spikelets, the transcription level of *TRI5* also was similar between the wild type and *osp24* mutant (Supplementary Fig. 3). Interestingly, the *OSP24, OSP25*, and *OSP44* genes all are located in the sub-telomeric regions (Fig. 1h), which may facilitate their rapid evolution[39]. Moreover, they were all specifically expressed or significantly up-regulated in infected wheat heads (Fig. 1i; Supplementary Table 3). These results indicate that the secreted Osp24, Osp25, and Osp44 proteins are important for the full virulence of *F. graminearum*, possibly by functioning as effectors during plant infection.

**The *osp24* mutant is defective in infectious growth in the rachis tissues**. Because the *osp24* deletion mutant was most significantly reduced in virulence, we selected it for further characterization. In infection assays with corn silks, tomatoes and Arabidopsis floral tissues, the *osp24* mutant was normal in virulence (Supplementary Fig. 4), suggesting a host-specific role of Osp24 during plant infection. To further characterize its function during wheat

infection, the *osp24* mutant was assayed for the formation of infection cushions and growth of invasive hyphae. When examined by scanning electron microscopy (SEM), abundant infection cushions were observed on wheat lemma inoculated with the *osp24* mutant at 2 days post-inoculation (dpi). In comparison with the wild type, the *osp24* mutant had no obvious defects in infection cushion formation (Fig. 2a). Deletion of *OSP24* also had no obvious effects on the initial plant penetration and development of invasive hyphae in lemma tissues at 2 dpi (Supplementary Fig. 5). However, extensive hyphal growth to the wild-type level was not observed in the rachis tissues of *osp24*-infected wheat heads at 5 dpi (Fig. 2b). When assayed by qPCR with genomic DNA isolated from infected wheat heads at 5 dpi, fungal biomass was significantly reduced in wheat heads inoculated with the *osp24* mutant in comparison with those inoculated with PH-1 (Fig. 2c). These results indicate that *OSP24* plays an important role in infectious growth and spreading in the wheat rachis tissues.

**The SP of Osp24 is required for its secretion and function**. The SP-*SUC2* construct was generated by cloning the 22-aa SP of

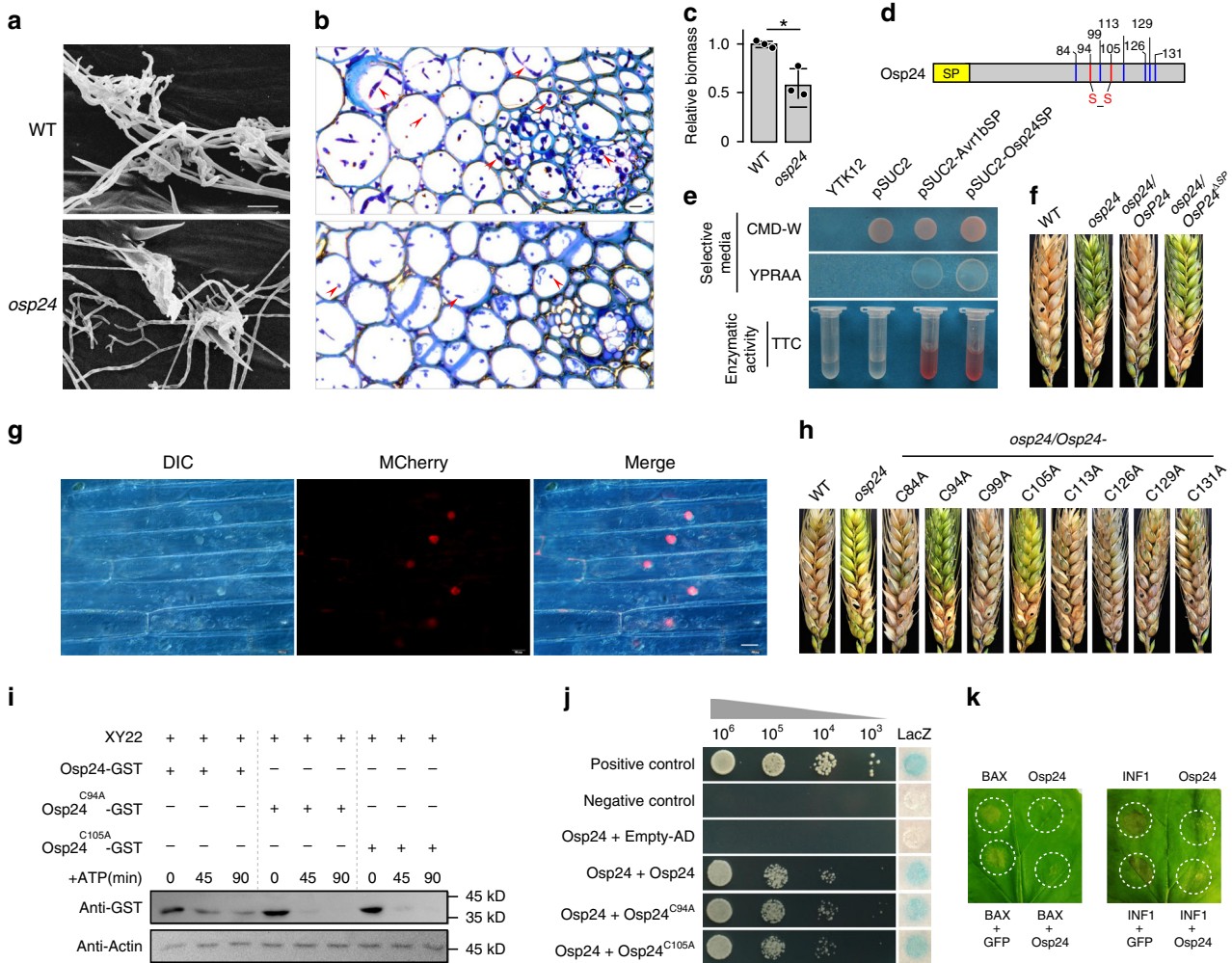

**Fig. 2 Functions of Osp24 and its signal peptide and cysteine residues. a** Infection cushions formed by the wild-type strain PH-1 (WT) and *osp24* deletion mutant on wheat lemma at 2 dpi were examined by SEM under ×800 amplification. Scale bar, 10 μm. **b**. Thick sections of infected wheat heads were examined for invasive hyphae (red arrowheads) in the rachis tissues at 5 dpi. Scale bar, 20 μm. **c** Relative biomass of *F. graminearum* in infected wheat heads at 5 dpi was determined by qPCR. Mean and standard deviation were estimated with data from three ($n = 3$) independent biological replicates (marked as black dots on the bar). The asterisk * indicates significant differences ($P = 0.05$) based on Bootstrap analysis. **d** The positions of eight cysteine residues in Osp24 and the predicted intra-molecular disulfide bond between C94 and C105. SP, signal peptide. **e** The yeast *suc2* mutant YTK12 and its transformants expressing the empty vector pSUC2 or vectors with the signal peptide from Osp24 and Avr1 (positive control) were assayed for growth on CMD-W or YPRAA plates and invertase activity in TTC medium. **f** Representative images of wheat heads infected with PH-1, *osp24* mutant, and the *osp24/OSP24* and *osp24/OSP24*^ΔSP transformants were photographed at 14 dpi. **g** Wheat coleoptiles were infected with PH-1 expressing the Osp24: mCherry:NLS construct and examined for mCherry signals in plant cells. Scale bar, 20 μm. **h** Representative images of wheat heads infected with PH-1, *osp24* mutant, and transformants of *osp24* expressing *OSP24* mutant alleles carrying the indicated C-to-A mutations were photographed at 14 dpi. **i** Western blots of mixtures of equal amounts of total proteins isolated from wheat heads of cultivar Xiaoyan 22 (XY22) and recombinant Osp24-GST, Osp24^C94A-GST, or Osp24^C105A–GST proteins incubated for the indicated times after the addition of 10 mM ATP were detected with an anti-GST antibody. Detection with an anti-actin antibody was used as a loading control. **j** Different concentrations (cells/ml) of yeast transformants expressing the indicated bait and prey constructs were assayed for growth on SD-Trp-Leu-His plates and LacZ activity. **k** Transient expression of Osp24 suppressed programmed cell death triggered by BAX or INF1. At the indicated spots, *N. benthamiana* leaves were infiltrated with Agrobacterium cells expressing GFP/ BAX/INF1 alone or infiltrated with BAX/INF1-expressing cells at 18 h after infiltration with GFP or Osp24 first. Representative leaves were photographed 5 days after infiltration.

Osp24 (Fig. 2d) into the vector pSUC2[40] and transformed into the yeast *suc2* mutant[41]. The resulting SP-SUC2 transformants were able to grow on YPRAA agar and had secreted invertase activity (Fig. 2e), indicating that the SP of Osp24 is functional in yeast.

To test its function in *F. graminearum*, we then generated the *OSP24* and *OSP24*^ΔSP constructs[42,43] and transformed them into the *osp24* mutant. The resulting *osp24/OSP24* transformants were normal in virulence. However, the *osp24/OSP24*^ΔSP transformants were similar to the *osp24* mutant in virulence on wheat heads

(Fig. 2f), indicating that the SP is essential for the complementation of *osp24* mutant by ectopic expression of *OSP24*. Therefore, the secretion of Osp24 is important for its function during plant infection.

**Osp24 is a cytoplasmic effector that is translocated into plant cells.** To determine whether Osp24 is delivered into plant cells, we generated the P_OSP24-*OSP24*-mCherry-NLS construct with the NLS from SV40 T-Antigen and transformed it into PH-1. In wheat seedlings infected with transformants expressing the

Osp24-mCherry-NLS construct, mCherry signals were observed in the nucleus in coleoptile cells at the inoculation sites (Fig. 2g), suggesting that Osp24 is a cytoplasmic effector that is translocated into plant cells.

**The C94 and C105 cysteine residues are important for the function of Osp24.** Similar to many other fungal effectors, Osp24 is a cysteine-rich protein with eight cysteine (C) residues (Fig. 2d). To determine their roles in Osp24 function, constructs of *OSP24* with individual Cs changed to alanine (A) were generated and transformed into the *osp24* mutant. Whereas mutations of six other cysteine residues had no effect on virulence, the *osp24/OSP24*C94A and *osp24/OSP24*C105A transformants, similar to the *osp24* mutant, were defective in wheat head infection (Fig. 2h), indicating that C94 and C105 are essential for the function of Osp24 proteins during pathogenesis.

When analyzed for the formation of intra-molecular disulfide bonds with the EDBCP tool[44], the disulfide bridge between C94 and C105 was predicted with the highest probability (Probability = 0.89774). To characterize their functions, we expressed and purified Osp24-, Osp24C94A-, and Osp24C105A- GST fusion proteins. Equal amounts of these recombinant proteins were co-incubated with total proteins isolated from wheat heads inoculated with PH-1 sampled at 3 dpi and assayed for their degradation by western blot analyses. In comparison with Osp24-GST proteins that were degraded gradually over time, the degradation of Osp24C94A and Osp24C105A GST fusion proteins occurred much more rapidly (Fig. 2i), indicating that Osp24 proteins may be less stable when C94 or C105 is changed to A. Therefore, the intra-molecular disulfide bond between C94 and C105 may be important for the folding and stability of Osp24 proteins.

When tested by yeast two-hybrid assays, the Osp24 protein was found to interact with itself (Fig. 2j), suggesting that Osp24 may form homodimers. To determine their roles in the formation of homodimers, we also generated the bait constructs of Osp24 carrying the C94A or C105A mutation. Similar to the Osp24–Osp24 interaction, Osp24 interacted with both Osp24C94A and Osp24C105A in yeast two-hybrid assays (Fig. 2j), indicating that the C-to-A mutation at C94 or C105 had no obvious effect on its intermolecular interaction. Therefore, C94 and C105 are likely only important for the formation of intramolecular disulfide bonds.

**Osp24 suppresses cell death and the expression of metabolism-related genes.** To characterize its role as an effector to suppress host immunity, we first assayed the effect of Osp24 on programmed cell death (PCD) in *Nicotiana benthamiana* induced by BAX, a mouse proapoptotic protein[45]. The $P_{CaMV\ 35S}$-Osp24$^{\Delta SP}$ construct was generated and transformed into *Agrobacterium tumefaciens*. In *N. benthamiana* leaves infiltrated with Agrobacterium expressing Osp24$^{\Delta SP}$ alone, no cell death was observed. Under the same conditions, PCD was observed on leaves infiltrated with cells expressing BAX. However, on leaves infiltrated with *A. tumefaciens* expressing Osp24$^{\Delta SP}$ 18 h before infiltration with cells expressing BAX, cell death was not observed when Osp24$^{\Delta SP}$ and BAX proteins were co-expressed (Fig. 2k, Supplementary Fig. 6), indicating suppression of BAX-induced cell death by Osp24. We also tested the effects of Osp24 on PCD triggered by INF1, a pathogen-associated molecular pattern (PAMP) from *Phytophthora infestans*[46]. In infiltration assays with *N. benthamiana* leaves, INF1-induced cell death also was suppressed by Osp24 (Fig. 2k).

To identify genes suppressed or induced by Osp24 during plant infection, we conducted RNA-seq analysis with infected wheat heads that were collected at 3 dpi. A total of 478 differentially expressed genes (DEGs) with at least two-fold changes were identified in wheat infected with the wild type and *osp24* mutant. Among them, 296 DEGs were up-regulated in wheat inoculated with the *osp24* mutant, suggesting that their expression may be suppressed by Osp24 during *F. graminearum* infection. FunCat-enrichment analysis showed that up-regulated DEGs were significantly enriched for genes functionally related to metabolism, information pathway, and perception and response to stimuli (Supplementary Fig. 7). Gene Ontology (GO)-enrichment analysis revealed that many GO categories associated with metabolism were significantly enriched, including oxidation–reduction, glycolytic, glycine catabolic, malate metabolic, peroxisome fission, chlorophyll biosynthetic, fructose 2,6-bisphosphate metabolic, fructose metabolic, L-serine metabolic, photosynthesis, and ATP synthesis coupled proton transport processes (Supplementary Data 1). Interestingly, 17 (~7%) of the up-regulated DEGs in *osp24*-infected wheat heads encode putative NBS-LRR proteins that are known to be involved in plant immunity against microbial pathogens[47]. Putative NBS-LRR genes were absent in the down-regulated DEGs. It is possible that some of these NBS-LRR genes with up-regulated expression in *osp24*-infected wheat heads may contribute to defense responses against *F. graminearum* infection.

In comparison with the wild type, only 54 and 83 fungal genes were up-regulated and down-regulated, respectively, in the *osp24* mutant during wheat infection. Because none of them are related to known virulence factors or putative effector genes (Supplementary Data 2), their altered expression levels may not contribute significantly to the defect of the *osp24* mutant in virulence.

**Osp24 interacts with wheat TaSnRK1α.** To further characterize its function during plant infection, we generated an Osp24 bait construct and screened a yeast two-hybrid library constructed with RNA isolated from infected wheat heads collected at 3 dpi. After screening over five coverage of this library, 16 putative Osp24-interacting clones (OICs) were identified (Supplementary Table 4). Sequencing analysis showed that five clones were the SNF1-related protein kinase gene TaSnRK1α[28] that is highly similar to Arabidopsis SnRK1. In Arabidopsis, SnRK1 is a central integrator of stress and energy signaling that regulates plant metabolism, growth, and immunity[48,49].

The wheat genome has three TaSnRK1α homoeologues (A, B, and D) that share the same AA sequence and have only 12–14 nucleotide differences in the coding regions[50]. Because all the five TnSnRK1α clones identified in the original yeast two-hybrid library screen were TaSnRK1α-A, we used TnSnRK1α-A for all the experiments related to TaSnRK1α unless otherwise stated. We then generated the full-length TaSnRK1α prey construct and confirmed its interaction with Osp24 in yeast two-hybrid assays (Fig. 3a). To verify their interaction in plant cells, the Osp24-nYFP and TaSnRK1α-cYFP fusion constructs were generated and co-expressed in *N. benthamiana* leaves by Agrobacterium infiltration. In epidermal cells of infiltrated tobacco leaves, YFP signals were observed in the nucleus but not in the cytoplasm (Fig. 3b), indicating the expression of Osp24-nYFP and TaSnRK1α-cYFP and their interaction in the nucleus of plant cells. We further verified the interaction between Osp24 and TaSnRK1α by in vitro pull-down assays. Recombinant Osp24-GST and TaSnRK1α-HIS proteins were purified from *Escherichia coli* and co-incubated with glutathione resins. In western blots of proteins eluted from glutathione resins, the TaSnRK1α-HIS band was detected (Fig. 3c), indicating its interaction with Osp24-GST in vitro.

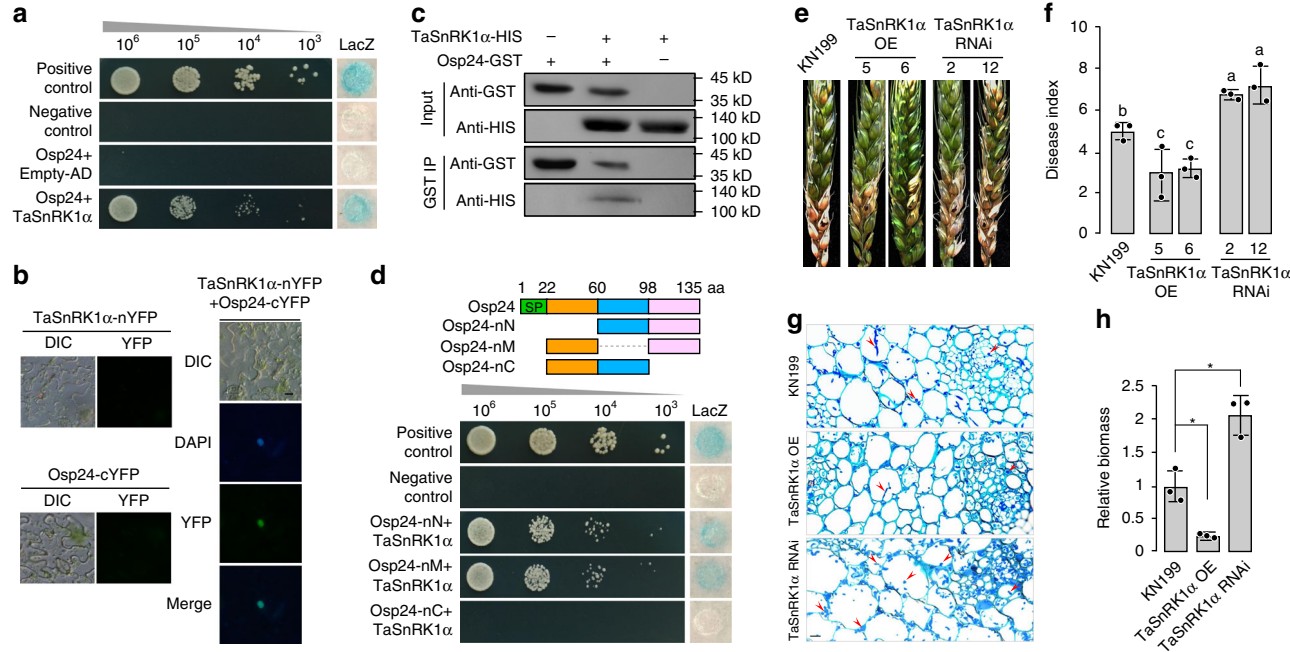

**Fig. 3 Interaction of Osp24 with TaSnRK1α and function of TaSnRK1α in wheat resistance against *F. graminearum*. a** Yeast two-hybrid assays to detect the interaction of Osp24 (Bait) with TaSnRK1α (Prey). Different concentrations of the labeled yeast transformants were assayed for growth on SD-Trp-Leu-His plates and LacZ activity. **b** BiFC assays for the interaction of Osp24 with TaSnRK1α. Leaves of *N. benthamiana* were agroinfiltrated with a mixture of *A. tumefaciens* strains expressing the Osp24-nYFP and TaSnRK1α-cYFP constructs. YFP signals were observed at 2 days post-agroinfiltration. Infiltration with Agrobacterium expressing the TaSnRK1α or Osp24 construct alone was used as the negative control. No YFP signals was observed in these negative controls. Scale bar, 20 μm. **c** Verification of the Osp24–TaSnRK1α interaction by GST pull down assays. Western blots of the marked protein mixtures (Input) or proteins co-purified with TaSnRK1α-HIS from these mixtures (GST IP) were detected with the anti-HIS and anti-GST antibodies. **d** Yeast transformants expressing the TaSnRK1α (prey) and marked full-length or truncated Osp24 (bait) were assayed for growth on SD-Trp-Leu-His plates and LacZ activities. SP signal peptide. **e** Representative wheat heads of cultivar KN199 and its transgenic lines expressing the TaSnRK1α overexpression (TaSnRK1α OE 5 and 6) or silencing (TaSnRK1α RNAi 2 and 12) construct were drop-inoculated with PH-1 and photographed at 7 dpi. **f** Mean and standard deviation (SD) of the disease index of PH-1 on the labeled wheat lines were estimated from three independent experiments (*n* = 3) with at least 10 infected wheat heads in each experiment. Different letters indicate significant differences based on ANOVA analysis followed by Duncan's multiple range test (*P* = 0.05). **g** Thick sections of the rachises of wheat heads of KN199 and its TaSnRK1α OE or TaSnRK1α RNAi transgenic line inoculated with PH-1 were examined for invasive hyphae (red arrowheads) at 5 dpi. Scale bar, 50 μm. **h** Relative biomass of *F. graminearum* was determined by qPCR in infected wheat heads of KN199 and labeled transgenic lines sampled at 5 dpi. Mean and standard deviation were estimated with data from three (*n* = 3) independent biological replicates (indicated with black dots on the bar). *Indicates significant differences (*P* = 0.05) based on Bootstrap analysis.

To determine the region required for its association with TaSnRK1α, three different fragments of Osp24 (Fig. 3d) were amplified and tested in yeast two-hybrid assays. Deletion of the C-terminal region (aa 99–135) of Osp24 eliminated its interaction with TaSnRK1α (Fig. 3d). However, deletion of the N-terminal (aa 1–60) or middle (aa 61–98) region had no obvious effect on the Osp24–TaSnRK1α interaction (Fig. 3d). Therefore, the C-terminal region of Osp24 is involved in its interaction with TaSnRK1α. In Agrobacterium infiltration assays, expression of Ops24 with a truncated C-terminal region failed to suppress BAX-induced or INF1-induced cell death in *N. benthamiana* (Supplementary Fig. 8). These results indicate that the C-terminal region of Osp24 is related to its function in PCD suppression, likely by interacting with SnRK1.

**TaSnRK1α contributes to resistance against *F. graminearum*.** To determine its function in resistance against *F. graminearum*, we first generated a TaSnRK1α RNAi construct and transformed it into wheat cultivar KN199[51], which is a cultivar amenable to efficient transformation but more tolerant to FHB than cultivar Xiaoyan 22. In transgenic lines, the expression level of TaSnRK1α was reduced approximately two-fold based on qRT-PCR assays (Supplementary Fig. 9a). In wheat head infection assays, transgenic lines with reduced TaSnRK1α expression displayed more

severe FHB symptoms than cultivar KN199 when inoculated with the wild-type strain PH-1 (Fig. 3e). The disease index of PH-1 was increased in lines expressing the TaSnRK1α RNAi construct in comparison with KN199 (Fig. 3f).

We also generated transgenic lines with the TaSnRK1α gene under the control of the CaMV 35S promoter[52]. Increased expression of TaSnRK1α (over 3-fold) was detected in five transgenic lines (Supplementary Fig. 9b). In wheat head infection assays, transgenic plants overexpressing TaSnRK1α were increased in resistance to *F. graminearum* (Fig. 3e). The disease index of PH-1 on TaSnRK1α overexpressing plants was decreased ~40% compared to that on KN199 (Fig. 3f).

Overexpression of TaSnRK1α also significantly reduced infectious growth and only limited invasive hyphae were observed in the rachis tissues of transgenic wheat plants overexpressing TaSnRK1α (Fig. 3g). When assayed by qPCR, less fungal biomass was detected in infected wheat heads in transgenic lines overexpressing TaSnRK1α than in KN199 (Fig. 3h). In contrast, more abundant infectious hyphae were observed in the rachis tissues (Fig. 3g), and more fungal biomass was detected by qPCR (Fig. 3h) in wheat heads of transgenic plants expressing the TaSnRK1α RNAi construct than in the control KN199 plants. Nevertheless, DON production in the diseased spikelets inoculated with PH-1 was similar between KN199 and TaSnRK1α

RNAi plants (Supplementary Fig. 10). These results indicate that overexpression of TaSnRK1α in wheat plants increased the resistance against *F. graminearum* and silencing of TaSnRK1α had the opposite effect.

**Osp24 accelerates TaSnRK1α degradation during infection**. To assay the effect of Osp24 on TaSnRK1α proteins in a cell-free degradation assay, equal amounts of TaSnRK1α-HIS recombinant proteins were mixed and co-incubated with total proteins isolated from wheat heads of cultivar Xiaoyan 22 inoculated with the wild-type strain PH-1 or *osp24* mutant (sampled at 3 dpi). In the presence of ATP, TaSnRK1α was degraded over time (45–90 min) in both co-incubation mixtures (Fig. 4a). However, the rate of TaSnRK1 degradation was reduced in co-incubation mixtures with proteins from *osp24*-infected wheat heads in comparison with those with proteins from PH-1-infected samples (Fig. 4a). These results suggest that Osp24 may accelerate TaSnRK1α degradation during infection. Under the same conditions, TaSnRK1α degradation was not observed in the co-incubation mixture of TaSnRK1α-HIS and Osp24-GST without proteins from infected wheat heads (Fig. 4b). However, the degradation of TaSnRK1α-HIS was observed in its co-incubation mixture with Osp24-GST and protein extracts from non-inoculated wheat heads (Fig. 4c), indicating that Osp24-stimulated degradation of TaSnRK1α is dependent on some components of wheat head protein extracts.

We also assayed the transcription level of TaSnRK1α during *F. graminearum* infection. TaSnRK1α transcription was not up-regulated in infected wheat heads in comparison with non-inoculated samples (Supplementary Fig. 11a). Furthermore, wheat plants inoculated with PH-1 or the *osp24* mutant had no obvious difference in the abundance of TaSnRK1α transcripts (Supplementary Fig. 11b). These results indicate that Osp24 does not affect the transcription of TaSnRK1α but may stimulate its degradation in infected wheat heads.

**Osp24 facilitates the interaction of TaSnRK1α with the ubiquitin-26S proteasome system**. In *Arabidopsis*, SnRK1 interacts with both the SCF ubiquitin ligase complex and 26S proteasome[53]. Osp24 may promote the degradation of TaSnRK1α through the ubiquitin-26S proteasome pathway[54] during fungal infection. To test this hypothesis, we first assayed the effect of proteasomal inhibitor MG132 on TaSnRK1α degradation. Even in the presence of Osp24, the degradation of TaSnRK1α in co-mixtures with proteins isolated from infected wheat heads was suppressed by MG132 (Fig. 4d), indicating the involvement of the 26S proteasome.

We then investigated the effects of Osp24 on the interaction between TaSnRK1α and the ubiquitin-26S proteasome by in vitro pull-down assays. TaSnRK1α-HIS proteins were co-incubated with total proteins isolated from wheat heads in the presence or absence of Osp24-GST for 30 min before mixing with anti-HIS beads. Proteins bound to anti-HIS beads were then eluted and assayed for the presence of SCF ubiquitin ligase complex and 26S proteasome with the anti-RBX1 and anti-20S proteasome antibodies[37], respectively. With or without the addition of Osp24-GST, both SCF ubiquitin ligase complex and 26S proteasome were detected in proteins co-immunoprecipitated with TaSnRK1α-HIS (Fig. 4e). However, the presence of Osp24 increased the amount of these proteins co-purified with TaSnRK1α-HIS (Fig. 4e). These results indicate that Osp24 may accelerate TaSnRK1α degradation by enhancing its association with the ubiquitin-26S proteasome system.

**The wheat orphan protein TaFROG competes with Osp24 in binding with TaSnRK1α**. TaSnRK1α is known to interact with a wheat orphan protein TaFROG, which is induced by DON[28]. In yeast two-hybrid assays, the C-terminal region of TaSnRK1α (267–499 aa) was found to interact with both Osp24 and TaFROG (Fig. 4f). In contrast, the N-terminal region of TaSnRK1α (1–266 aa) was dispensable for its interaction with Osp24 or TaFROG. The direct interaction between Osp24 and TaFROG was not detected in yeast two-hybrid assays (Supplementary Fig. 12).

Because both Osp24 and TaFROG interacted with the same region of TaSnRK1α, it is possible that they may compete with each other for interacting with TaSnRK1α. To test this hypothesis, TaSnRK1α-HIS, Osp24-GST, and TaFROG-S-tag fusion proteins were purified and used in in vitro pull-down assays. When equal amounts of TaSnRK1α-HIS and Osp24-GST proteins were co-incubated and mixed with anti-HIS beads, abundant TaSnRK1α proteins were detected in proteins co-purified with Osp24-GST. With the addition of increasing concentrations of TaFROG proteins to the TaSnRK1α–Osp24 protein mixtures, the amount of TaSnRK1α proteins co-purified with Osp24-GST was gradually reduced in a concentration-dependent manner (Fig. 4g), indicating the competition between TaFROG and Osp24 in binding with TaSnRK1α.

**Overexpressing TaFROG enhances FHB resistance by stabilizing TaSnRK1α**. To determine the role of TaFROG in regulating TaSnRK1α stability during *F. graminearum* infection, we generated transgenic wheat plants in which the TaFROG gene was overexpressed with the CaMV 35S promoter. When assayed by qRT-PCR, the expression level of TaFROG was increased over 100-fold in five transgenic lines (Supplementary Fig. 13). In wheat head infection assays, fewer spikelets developed FHB symptoms in the TaFROG overexpression transgenic lines than the control KN199 plants (Fig. 4h, i), indicating an increase in resistance against *F. graminearum* by TaFROG overexpression, which is consistent with an earlier report[28]. We then conducted cell-free degradation assays with equal amounts of TaSnRK1α-HIS recombinant proteins co-incubated with total proteins isolated from wheats heads of control KN199 and TaFROG overexpression plants infected with PH-1. Although TaSnRK1α degradation was observed in all the reaction mixtures, the degradation rate was significantly reduced in co-incubation mixtures with proteins isolated from transgenic lines over-expressing TaFROG in comparison with those from KN199 plants (Fig. 4j), confirming that overexpression of TaFROG increases the stability of TaSnRK1α. In the same degradation assays, the rate of TaSnRK1α degradation was similar in co-incubation mixtures with proteins isolated from wheat heads infected with PH-1 or the *osp24* mutant (Supplementary Fig. 14). Therefore, overexpression of TaFROG may enable its effective protection of TaSnRK1α against Osp24 binding and degradation.

TaFROG may compete with Osp24 in binding with TaSnRK1α and reduce its degradation via the ubiquitin-26S proteasome system. To test this hypothesis, TaSnRK1α-HIS proteins were mixed with total proteins isolated from wheat heads infected with PH-1 and anti-HIS beads. Western blots of proteins eluted from anti-HIS beads were then detected with the anti-RBX1 and anti-20s proteasome antibodies. The density of both SCF ubiquitin ligase complex and 26S proteasome bands was weaker in samples with proteins isolated from transgenic plants overexpressing TaFROG than those with proteins from KN199 (Fig. 4k). These results indicate that overexpression of TaFROG may reduce the interaction of TaSnRK1α with the ubiquitin-26S proteasome

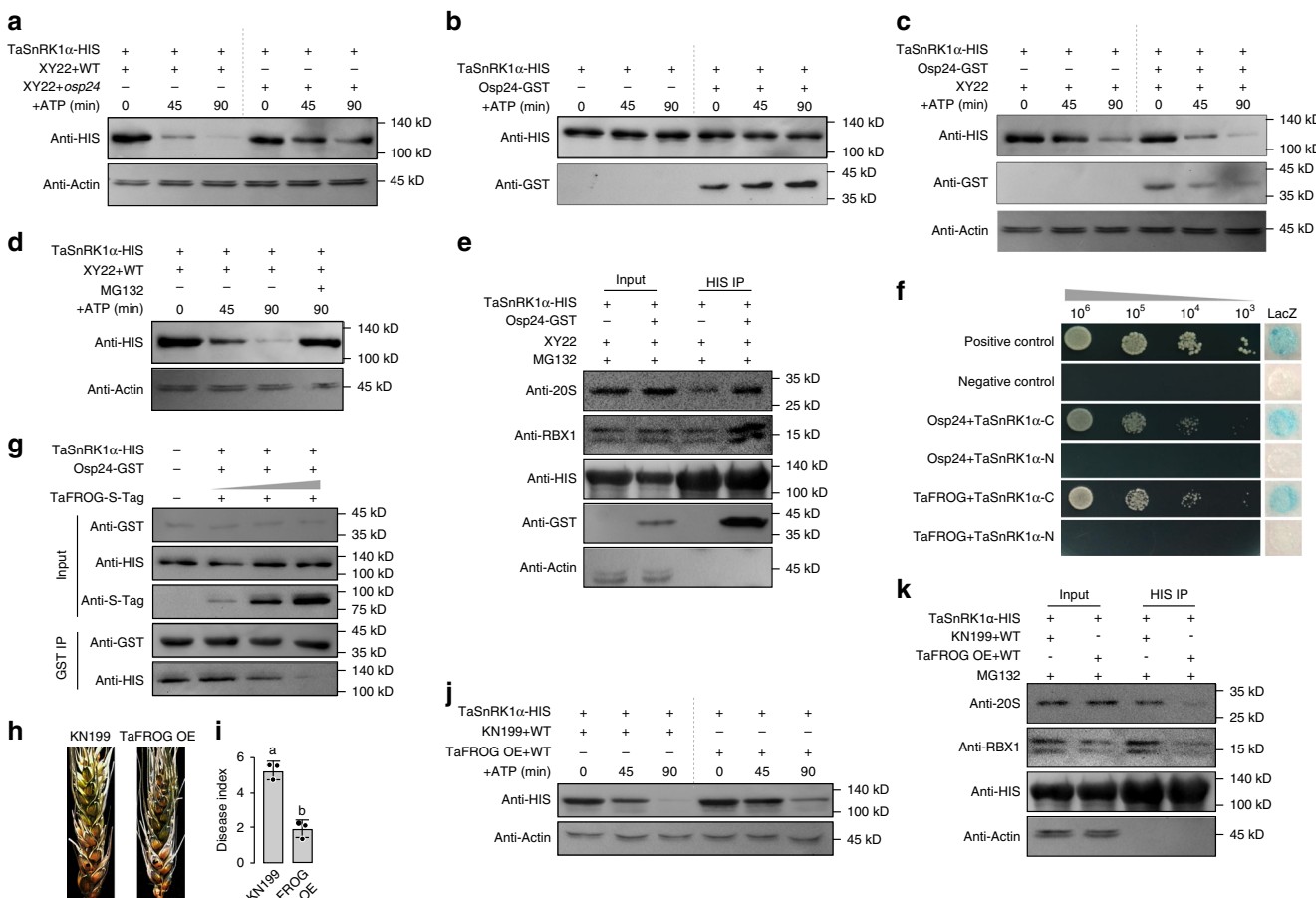

**Fig. 4 Roles of Osp24 and TaFROG in modulating TaSnRK1α stability. a** Western blots of the mixtures of equal amount of TaSnRK1α-HIS and total proteins isolated from wheat heads of cultivar Xiaoyan22 inoculated with the wild type PH-1 (XY22 + WT) or osp24 mutant (XY22 + osp24) incubated for the indicated times after the addition of 10 mM ATP were detected with the anti-HIS or anti-actin (loading control) antibody. **b** Western blots of the mixtures of equal amount of TaSnRK1α-HIS and Osp24-GST incubated for the indicated times were detected with an anti-HIS or anti-GST antibody. Degradation of TaSnRK1α-HIS by Osp24 was not observed. **c** Recombinant TaSnRK1α-HIS and Osp24-GST proteins were incubated with equal amounts of total proteins isolated from wheat head (XY22) in the presence of 10 mM ATP and used for western blot analyses with the indicated antibodies. **d** Degradation of TaSnRK1α by total proteins isolated from wheat head inoculated with PH-1 (XY22 + PH-1) was inhibited by proteasome inhibitor MG132. **e** Western blots of the indicated protein mixtures (Input) or proteins co-purified with TaSnRK1α-HIS from these protein mixtures (HIS IP) were detected with the anti-20S, anti-RBX1, anti-HIS, and anti-GST antibodies. **f** Yeast transformants expressing the indicated Osp24 or TaFROG bait construct and prey constructs of the N-terminal (1–266 aa) or C-terminal (267–499 aa) region of TaSnRK1α (TaSnRK1α-N and TaSnRK1α-C) were assayed for growth on SD-Trp-Leu-His plates and LacZ activity. **g** Western blots of the indicated protein mixtures (Input) or proteins co-purified with Osp24-GST from these mixtures (GST IP) were detected with the anti-GST, anti-HIS, or anti-S-tag antibody. The amount of TaSnRK1α-HIS proteins co-immunoprecipitated with Osp24-GST was reduced by the addition of increasing concentrations of TaFROG-S-tag proteins. **h** Representative wheat heads of cultivar KN199 and its TaFROG overexpressing transgenic line (TaFROG OE) infected with PH-1 were photographed at 8 dpi. **i** Mean and standard deviation (SD) of the disease index of PH-1 on the labeled wheat lines were estimated from three independent experiments ($n = 3$) with at least 10 infected wheat heads in each replicate. Different letters indicate significant differences based on ANOVA analysis followed by Duncan's multiple range test ($P = 0.05$). **j** Cell-free degradation assays with recombinant TaSnRK1α-HIS incubated with equal amounts of total proteins isolated from wheat heads of KN199 and TaFROG OE transgenic lines infected by PH-1 for the indicated time after addition of 10 mM ATP. **k** Western blots of the indicated protein mixtures (Input) or proteins co-purified with TaSnRK1α-HIS from these protein mixtures (HIS IP) were detected with the anti-20S, anti-RBX1, and anti-HIS antibodies. For **a**, **c**, **d**, **e**, **j**, and **k**, detection with an anti-actin antibody was used as the control.

system in these transgenic plants, possibly by competing with Osp24.

## Discussion

Like many other plant pathogenic fungi, *F. graminearum* may use secreted proteins or effectors to suppress plant defense responses and its genome contains hundreds of orphan genes that are specifically expressed or highly up-regulated during plant infection. In this study, a total of 50 secretory proteins unique to *F. graminearum* were identified and functionally characterized. Among them, *OSP*, *OSP25*, and *OSP44*, were found to be

important for virulence. They all have typical features of fungal effectors[4,5] and are in the fast-evolving telomeric regions[55]. The location of these three putative effector genes in the telomeric regions may allow the rapid gain and loss or translocation to supernumerary chromosomes as effector reservoirs[39,56] although supernumerary chromosomes have not been reported in the few sequenced *F. graminearum* strains.

*OSP24* was selected for further characterization because the *osp24* deletion mutant had the most significant reduction in virulence. Although dispensable for vegetative growth, reproduction, and initial penetration, *OSP24* is important for infectious

growth in the rachis tissues in infected wheat heads. Osp24 lacks any conserved domain but is a cysteine-rich protein. Two of the eight cysteine residues, C94 and C105, were found to be important for the function of Osp24 in plant infection. In *Sclerotinia sclerotiorum*, two cysteine residues of effector SsSSVP1 are essential for the formation of a homo-dimer and its interaction with the host target[57]. Although Osp24 may form homo dimers, C94 and C105 were dispensable for the Osp24–Osp24 interaction in yeast two-hybrid assays. In fact, these two cysteine residues were predicted to form an intra-molecular disulfide bond. In *Stagonospora nodorum*, cysteine residues of effector SnTox1 may form multiple disulfide bonds to resist degradation[58]. In *F. graminearum*, the formation of a disulfide bond between C94 and C105 may be important for the proper folding and stability of Osp24 in infected wheat tissues.

Osp24 is a cytoplasmic effector that is translocated into plant cells during infection. In *M. oryzae*, a number of cytoplasmic effectors are accumulated in the BIC before being delivered into plant cells[34]. It is possible that Osp24 is translocated into wheat cells through BIC-like structures that may be formed by *F. graminearum*. Osp24 strongly interacted with TaSnRK1α in yeast two-hybrid assays and their interactions were confirmed by BiFC and in vitro pull-down assays. The highly conserved SnRK1 kinases function as metabolic regulators of energy homeostasis and are important for development and stress responses in plants[48,59]. SnRK1 kinases also are involved in regulating plant immunity and known to be targeted by viral and bacterial effector proteins[60–63]. Although SnRK1 is likely a conserved target for fungal pathogens, interactions between fungal secretory proteins and plant SnRK1 kinases have not been reported. In rice, the expression of OsSnRK1α was reported to be associated with disease resistance against *M. oryzae*, *Cochliobolus miyabeanus*, and *Rhizoctonia solani*[64,65] but its exact role in disease resistance has not yet been characterized. In wheat, the kinase activity of TaSnRK1α was increased in the presence of DON produced by *F. graminearum*[50]. In this study, we showed that overexpression of TaSnRK1α in wheat increased resistance against *F. graminearum* but transgenic plants expressing the SnRK1 silencing construct were more susceptible. It is likely that TaSnRK1α plays an important role in regulating resistance responses to *F. graminearum* and is a target of effector protein Osp24 secreted by the pathogen during infection.

The interaction of Osp24 with TaSnRK1α likely results in its degradation in infected wheat heads because the degradation of TaSnRK1α was accelerated in the presence of Osp24 in cell-free degradation assays. In Arabidopsis, the degradation of SnRK1 kinases is commonly associated with sumoylation and ubiquity-lation[66]. The association of SnRK1 kinases with SCF ubiquitin ligases complex and components of the 26S proteasome also have been well documented[53]. In this study, we showed that treatments with MG132, an inhibitor of the 26S proteasome[67], reduced Osp24-mediated TaSnRK1α degradation. Furthermore, we showed that Osp24 increased the association of TaSnRK1α with the SCF ubiquitin ligase complex and 26S proteasome, indicating that the stimulation of TaSnRK1α degradation by Osp24 likely occurs through the ubiquitination-26S proteasomal pathway[67]. Fungal effectors are known to target the ubiquitin-proteasome pathway in other pathosystems. Whereas effectors Pit2 of *U. maydis* and Avr2 of *Cladosporium fulvum* inhibit host proteases that are required for basal defense[68,69], AvrPiz-t of *M. oryzae* suppresses the RING E3 ubiquitin ligase APIP6 in plant cells[70]. Our results suggest that Osp24 is secreted by *F. graminearum* as an effector interacting with TaSnRK1α to stimulate its ubiquitination and proteasomal degradation, which in turn negatively impacts resistance responses to fungal infection in wheat plants.

Transient expression of Osp24 but not Osp24 truncated of its C-terminal region suppressed BAX-induced or INF1-induced cell death in *N. benthamiana*. Because the C-terminal region of Osp24 mediates its interaction with TaSnRK1α, it is possible that Osp24 suppresses PCD by targeting *N. benthamiana* SnRK1. The role of SnRK1 in PCD suppression was also reported in Pepper. Silencing of the pepper SnRK1 transcript resulted in a significant reduction of hypersensitive response (HR) that is elicited by protein AvrBs1 from *Xanthomonas campestris* pv. *vesicatoria* (*Xcv*)[62]. The YopJ effector homolog AvrBsT from *Xcv* targeted SnRK1 to suppress AvrBs1-induced plant immunity[62].

In *F. graminearum*, DON is not essential for the initial infection but plays a critical role in the spreading of infectious growth via the rachis in infected wheat heads. However, DON is not important for infecting Arabidopsis floral tissues[71], likely because infectious growth can spread by hyphae grown on the surface of floral tissues. Interestingly, the expression of TaFROG that encodes a *Pooideae*-specific orphan protein was induced by DON treatment. TaFROG interacts with TaSnRK1α, and over-expression of TaFROG increased resistance against *F. graminearum*[28]. In this study, we showed that Osp24 and TaFROG interacted with the same region of TaSnRK1α. Furthermore, we showed that their binding with TaSnRK1α was competitive and the binding of TaFROG with TaSnRK1α increased its stability (Fig. 5). It is likely that TaFROG functions in defense against *F. graminearum* by protecting TaSnRK1α against Osp24-mediated degradation. A recent report showed that TaFROG also interacts with TaNACL-D1, a *Poaceae*-divergent NAC transcription factor with its NAC C-terminal region specific to the *Triticeae*, to enhance FHB resistance independent of its interaction with TaSnRK1α[29]. It is possible that Osp24 also interacts with TaNACL-D1 to suppress defense against FHB. Osp24 and TaFROG are orphan proteins in the pathogens and hosts, respectively, and each may be subjected to co-evolution during fungal–plant interactions. To our knowledge, the active adoption of competing orphan proteins in both fungal pathogen and plant hosts has not been reported in other pathosystems. Expressing engineered TaFROG alleles with stronger interactions with TaSnRK1α or using the host-induced gene silencing (HIGS) approach to silence *OSP24* may improve resistance against *F. graminearum* without yield penalties because they encode orphan proteins specifically expressed during infection.

## Methods

**Identification of OSPs in *F. graminearum*.** The genome and predicted protein sequences of *F. graminearum*, *F. verticillioides*, and *F. oxysporum*, were downloaded from the Broad Institute website (ftp://ftp.broadinstitute.org/pub/annotation/fungi/fusarium/). To identify unique genes, protein sequences of *F. graminearum* were first used as the queries to search against the predicted proteomes of *F. verticillioides* and *F. oxysporum* by BLASTp. Sequences of the *F. graminearum* proteins also were used to search against the genome sequences of *F. verticillioides* and *F. oxysporum* by tBLASTn for possible genes that might not be predicted by automated annotation. (This study was initiated in 2010 when only the genome sequences of *F. verticillioides* and *F. oxysporum* were publicly available as the most closely related species of *F. graminearum*.) All protein sequences of *F. graminearum* without homologs in these searches (*E* value cutoff of 1e−5) were extracted and then used as queries to search against NCBI number database (excluding *F. graminearum* sequences) by BLASTp. The proteins without detectable homologs in these searches (*E* value cut off of 1e−5) were considered as orphan proteins of *F. graminearum*. Some of them are not unique to *F. graminearum* because they have homologs in the genome sequences of other *Fusarium* species that late became available in the public domain. Therefore, these sequences were described as orphan genes in *F. graminearum* and its close-related species. These predicted orphan proteins were further analyzed with SignalP 3.0 (http://www.cbs.dtu.dk/services/SignalP-3.0/) to identify secretory proteins.

**Culture conditions and fungal transformation.** The wild-type *F. graminearum* strain PH-1[33] and deletion mutants of *OSP* genes were routinely cultured on potato dextrose agar (PDA) plates at 25 °C. PDA cultures grown at 25 °C were used for measuring growth rate or colony morphology. Conidiation was assayed with

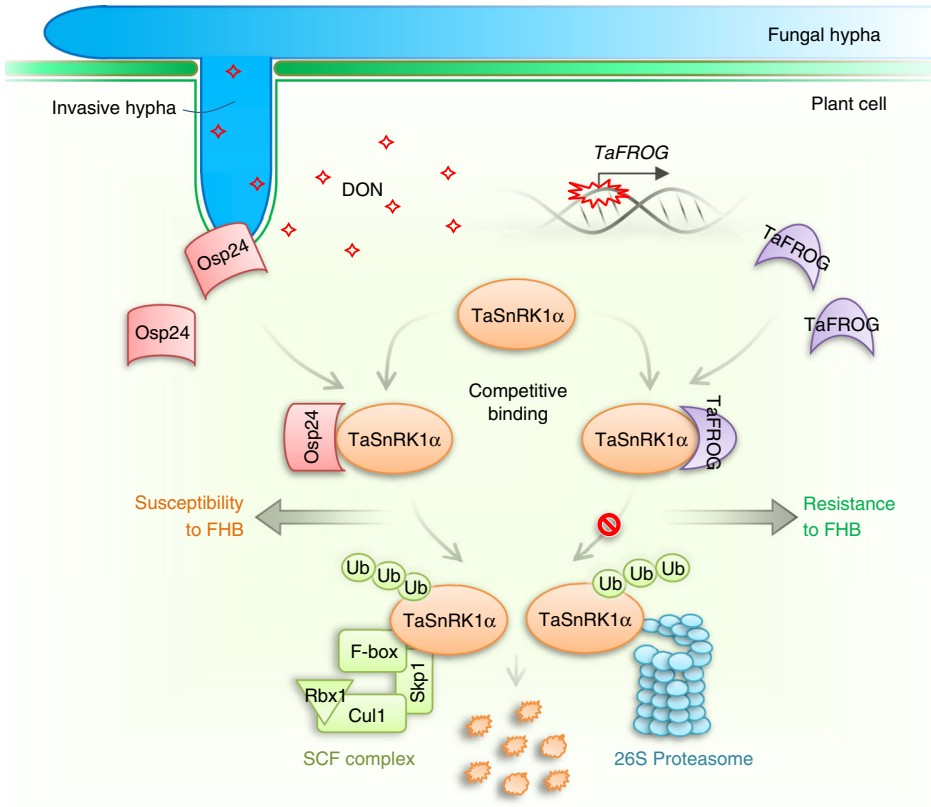

**Fig. 5 A schematic summary of the roles of two orphan proteins from wheat and *Fusarium graminearum* during fungal–plant interactions.** The cytoplasmic effector Osp24 secreted by invasive hyphae that develop in infected wheat tissues are translocated into plant cells. It interacts with TaSnRK1α, a protein kinase activated by DON and important for resistance against *F. graminearum* in wheat. The association of Osp24 with TaSnRK1α stimulates its degradation by the SCF ubiquitin ligase complex and 26S proteasome in infected plant cells, resulting in increased susceptibility to Fusarium head blight (FHB). However, DON produced by the fungal pathogen during infection induces the expression of TaFROG in wheat heads. TaFROG, a *Pooideae*-specific orphan protein, competes with Osp24 for binding with TaSnRK1α and increases its stability. Protection by TaFROG from proteasome degradation and activation by DON enhance the functions of TaSnRK1α in regulating defense responses against *F. graminearum* infection.

5-day-old liquid carboxymethyl cellulose (CMC) medium. For assaying defects in sexual reproduction, aerial hyphae of 5-day-old carrot agar cultures were pressed down with sterile 0.1% Tween 20 and then incubated at 25 °C under black light. Perithecium formation was examined 7 days after induction for sexual reproduction[17,72].

Gene replacement constructs were generated with the split-marker approach and transformed into protoplasts of PH-1[17,20]. For each *OSP* gene, at least two independent gene replacement mutants were identified. For complementation assays, the entire *OSP24* gene with its native promoter was cloned into plasmid pFL2 by gap repair[42,43] and transformed into the *osp24* mutant. The *osp24/OSP24* transformants were identified by PCR and assayed for phenotype complementation[17]. All the primers used in this study are described in Supplementary Data 3.

**Infection assays with flowering wheat heads**. Conidia were harvested from 5-day-old CMC cultures and re-suspended to 10^5 spores/ml in sterile distilled water (DDW). Flowering heads of 6-week-old wheat plants of cultivar Xiaoyan 22[73], Zhoumai 36[38], or KN199 were inoculated with 10 μl of conidium suspensions at the fifth spikelet from the base[72,74]. Inoculated wheat heads were capped with a plastic bag for 48 h to keep the moisture. Infected wheat heads were examined for diseased spikelets at 7 or 14 dpi to estimate the disease index (number of diseased spikeletsper head)[75]. Mean and standard deviation of the disease index were estimated with data from three independent replicates with at least 10 wheat heads examined in each replicate. DON production in the inoculated spikelets sampled at 14 dpi was assayed by GCMS-QP2010 system with AOC-20i autoinjector (Shimadzu Co. Japan). To assay infection cushion formation, infected lemmas were sampled at 2 dpi, fixed with 4% (vol/vol) glutaraldehyde, and coated with gold–palladium before examination with a JEOL 6360 scanning electron microscope (Jeol Ltd., Japan)[14,20,76]. For assaying infectious growth, infected rachis tissues were embedded in Spurr resin after fixation with 3% glutaraldehyde and dehydration in graded series of 30–100% of ethanol before being sectioned[77]. Thick sections were then prepared and stained with 0.5% (wt/vol) toluidine blue before examination with an Olympus BX-53 microscope. Differences between the wild-type and mutant strains in infection cushion formation and infectious growth were

determined with results from at least three independent replicates. Corn silks inoculated with culture blocks (5 mm) of *F. graminearum* were examined for discoloration at 7 dpi[78]. For infection assays with tomatoes, each fruit was injected with 10 μl of conidium suspensions after surface sterilization. Inoculated tomatoes were examined for tissue maceration after incubation at 25 °C in dark for 7 days[79]. For infection assays with Arabidopsis, flowers were sprayed with conidium suspensions of *F. graminearum* (10^5 spores/ml). Inoculated plants were cultured in plastic propagators to keep the moisture and examined for necrosis in floral tissues at 7 dpi[80].

**RNA-seq analysis**. The RNA-seq reads of wheat heads infected with PH-1 and *osp24* mutant were mapped to the reference genome of PH-1 via hisat2[81]. The expression level of each gene was counted with featureCounts[82]. DEGs were identified by edgeRun[83]. Functional Catalog (FunCat)[84] and GO-enrichment analyses were performed with custom script (https://github.com/xulab-nwafu/funcat) and Blast2GO[85], respectively.

**Assays for the function of the SP of Osp24**. The predicted SP (22 aa) of Osp24 (SP[22]) was cloned into the pSUC2 vector[86] that carries the yeast *SUC2* gene deleted of its SP sequence. The resulting SP[22]-*SUC2* construct was transformed into the yeast *suc2* mutant YTK12[87] and assayed for growth on CMD-W (0.67% yeast nitrogen base without AAs, 0.075% tryptophan dropout supplement, 2% sucrose, 0.1% glucose, and 2% agar) and YPRAA medium plate (1% yeast extract, 2% peptone, 2% raffinose, and 2 μg/ml antimicyn A)[88]. Transformants of YTK12 carrying the empty pSUC vector or pSUC2-Avr1b[SP89] were used as the negative and positive controls, respectively. The invertase enzymatic activity was detected by the reduction of 2,3,5-triphenyltetrazolium chloride (TTC) to insoluble red colored 1,3,5-triphenylformazan (TPF).

**Assays for the suppression of BAX-induced or INF1-induced cell death by Osp24**. P_CaMV 35S-Osp24^ΔSP was cloned into pGR106 and transformed into *A. tumefaciens* strain GV3101[90] expressing the BAX and INF1 constructs[91]. For infiltration assays with *N. benthamiana* leaves, *A. tumefaciens* cells were

resuspended to $OD_{600}$ of 0.8 in infiltration solution (10 mM MES, 10 mM $MgCl_2$, and 150 μM acetosyringone)[90]. At 18 h after the initial infiltration of tobacco leaves with *A. tumefaciens* transformant carrying the $P_{CaMV\ 35S}$-Osp24$^{\Delta SP}$ construct, the same sites were infiltrated with cells carrying GFP, BAX, or INF1 constructs[40]. Plant cell death symptoms were examined 5 days after infiltration with cells expressing BAX or INF1. *A. tumefaciens* cells carrying the GFP vector[40] were used as the negative control. Each infiltration experiment was repeated at least three times with a minimum of three leaves tested. The expression of BAX and Osp24 in *N. benthamiana* leaves were assayed by western blot analysis with the anti-BAX and anti-GFP antibodies.

**Assays for the localization of Osp24 during plant infection.** The Osp24-mCherry-NLS fusion construct under the control of its native promoter was generated by overlapping PCR with the NLS sequence from simian virus large T-antigen[92] and transformed into the wild-type stain PH-1. Conidium suspensions ($10^5$ spores/ml) of the resulting transformants were used for infection assays with wheat seedlings[15]. Infectious hyphae and mCherry signals in plant tissues were examined at 2 dpi with a Nikon A1 microscope at excitation/emission wavelengths 543 nm/560–615 nm.

**Yeast two-hybrid assays.** For library construction, RNA was isolated from wheat heads of Xiaoyan 22 inoculated with PH-1 and sampled at 3 dpi. The yeast two-hybrid library of 6 million primary clones was constructed with vector pGADT7 (Takara Bio, Japan) by OEBiotech (Shanghai, China). For library screening, the Osp24$^{\Delta SP}$ bait construct was generated with vector pGBKT7 (Takara Bio, Japan). Yeast colonies that grew on SD-Trp-Leu-His-Ade and had β-galactosidase activity were isolated as putative OICs. After sequencing with primer T7, wheat genes corresponding to the inserts in these clones were identified by Blast searches. To directly assay their interactions, full-length cDNAs or fragments of TaSnRK1α, Osp24, and TaFROG were amplified and cloned into pGADT7 or pGBKT7. The resulting bait and prey constructs of TaSnRK1α, Osp24, and TaFROG were transformed in pairs into yeast strain AH109. The Leu$^+$ and Trp$^+$ transformants were isolated and assayed for growth on SD-Trp-Leu-His medium and galactosidase activities in filter lift assays[78].

**BiFC assays for the TaSnRK1α–Osp24 interaction.** The TaSnRK1α and Osp24$^{\Delta SP}$ fragments were cloned into the BiFC vectors pSPYNE-35S and pSPYCE-35S[93], respectively. The resulting Osp24-nYFP and TaSnRK1α-cYFP constructs were transformed into *A. tumefaciens* strain GV3101. Leaves of *N. benthamiana* were then infiltrated with Agrobacterium cells expressing Osp24-nYFP and/or TaSnRK1α-cYFP as described above[91]. Two-days after infiltration, fluorescence signals were examined with a Nikon A1 microscope. Nuclei were stained with 4,6-diamidino-2-phenylindole (DAPI).

**In vitro pull-down assays.** Osp24, TaSnRK1α, and TaFROG cDNA fragments were amplified and cloned into pGEX4T1[94] and pCold[95] for purification of Osp24-GST, TaSnRK1α-His, and TaFROG-S recombinant proteins[96]. To assay their interactions, equal amounts of Osp24-GST and TaSnRK1α-HIS fusion proteins were incubated at 4 °C for 2 h and mixed with glutathione or Ni-NTA resins (GenScript, China) for affinity purification[97,98]. The presence of Osp24-GST and TaSnRK1α-HIS in proteins eluted from Ni-NTA resins was detected by western blot analysis with the anti-HIS (1:5000 dilution, CW0286, CoWin Bioscience Co., China) and anti-GST (1:5000 dilution, CW0084, CoWin Bioscience Co., China) antibodies. For assaying competition between Osp24 and TaFROG, equal amounts of Osp24-GST and TaSnRK1α-HIS fusion proteins were co-incubated with varying amounts of TaFROG-S-tag proteins (1×, 2×, and 4×) before mixing with Ni-NTA resins. To assay the interaction between TaSnRK1α and the ubiquitin-26S proteasome system, total proteins were isolated from wheat heads[20,78] and co-incubated with TaSnRK1α-HIS for 30 min before mixing with Ni-NTA resins in the presence of 50 μM proteasome inhibitor MG132[67]. Western blots of proteins eluted from the Ni-NTA resins were detected with the anti-RBX1 (1:1000 dilution, ab133565, Abcam, UK) and anti-20s (1:1000 dilution, ab22674, Abcam, UK) antibodies for the presence of RBX1 protein and 20S proteasome[37].

**Cell-free protein degradation assays.** Total proteins isolated from wheat heads[20,78] and TaSnRK1α-HIS recombinant proteins were used for the cell-free protein degradation assays[99,100]. In brief, a final concentration of 10 mM ATP was added to the mixture of equal amounts of TaSnRK1α-HIS and crude protein extract isolated from infected wheat heads[100]. Detection with an anti-actin antibody (1:1000 dilution, BE0027, Easybio, China) is used as a loading control. To inhibit the 26S proteasome activity, a final concentration of 50 μM MG132 was added to the reaction mixtures[67]. The degradation reaction was stopped after incubation at 25 °C for 0, 45, and 90 min by boiling for 5 min in SDS sample buffer. Western blots of these reaction mixtures were detected for the remaining TaSnRK1α-HIS proteins with an anti-HIS antibody (1:5000 dilution, CW0286, CoWin Bioscience Co., China). Each experiment was repeated at least three times.

**Generation of transgenic wheat plants.** For overexpression, the full-length TaSnRK1α ORF was cloned into the pANIC-5E vector behind the 35S promoter[101]. For silencing, a 107-bp fragment of TaSnRK1α was amplified and cloned into vector pANIC-7E in both antisense and sense orientations by Gateway cloning[101]. The resulting constructs were transformed into immature embryos of wheat cultivar KN199 by particle bombardment[32] at Genovo Bio (Tianjin, China). Transgenic plants resistant to BASTA were verified by PCR for carrying transforming constructs with DNA isolated from leaves of T0 plants. The expression level of TaSnRK1α was assayed by qRT-PCR with RNA isolated from wheat heads of selected T1 and T2 plants. Transgenic lines with over 2-fold increase or reduction of TaSnRK1α expression were selected for infection assays with *F. graminearum* as described above[20]. Similar approaches were used to generate TaFROG overexpression constructs and transgenic plants. To quantify fungal biomass in infected plant tissues, genomic DNA was extracted from infected wheat heads sampled at 5 dpi and used for qPCR assays with primers specific for the wheat GAPDH[102] and *F. graminearum* CHS5 genes[103]. Results from three independent biological replicates were used to estimate the ratio of wheat and *F. graminearum* biomasses.

**Reporting summary.** Further information on research design is available in the Nature Research Reporting Summary linked to this article.

## Data availability
Data supporting the major findings of this work are available within the paper and its Supplementary Information files. RNA-seq data were uploaded to the NCBI Sequence Read Archive under the accessions SRR12342821–SRR12342826. Any other supporting data is available from the corresponding authors upon request. Source data are provided with this paper.

## Code availability
The custom script for FunCat enrichment analysis is available from GitHub (https://github.com/xulab-nwafu/funcat).

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

## Acknowledgements

We thank Zhe Tang, Ping Xiang, Jiangang Kang, Gang Niu, and Guoyun Zhang for technical supports, Drs. Xue Zhang and Guanghui Wang for fruitful discussions, and Dr. Larry Dunkle at Purdue University for critical reading of this manuscript. We also thank Drs. Zhensheng Kang and Xiaojie Wang at Northwest A&F University for providing plasmids used in this study. This work was supported by grants from National Natural Science Foundation of China (Nos. 31671981, 31601587, and 31701747), New Star of Youth Science and Technology of Shaanxi Province (2018KJXX-068), Tang Scholar and USWBSI.

## Author contributions

C.J., J.-R.X., and H.Q.L. designed the experiments and wrote the paper. C.J., R.N.H., Y.Y., S.J.Z., Q.Z., W.W., M.Y., G.R.Z., and P.P.H. performed the experiments. C.J., H.Q.L., and Q.H.W. analyzed the data.

## Competing interests

The authors declare no competing interests.
