## [Peer Review File · Nature Communications]

Reviewers' comments:

Reviewer #1 (Remarks to the Author):

Overall this is an interesting manuscript which includes a wide range of different experiments. However, in many sections there are serious problems which the authors need to fully address.

Lineage specific adaptation

Lines 45 – 48 The authors hypothesis is that orphan genes are thought to play important roles in lineage-specific adaptation. Plant pathogenic fungi may evolve novel orphan genes to facilitate host adaptation and infection. Fungal effectors are good examples of orphan genes that have evolved for plant infection as many of them lack homologs in closely-related species.

But in this manuscript there is no experimental evidence provided that the orphan gene *Osp24* provides a lineage specific response because only the host species, wheat has been tested.

Bioinformatics analyses

The bioinformatics analysis done to reveal that the 50 selected 'orphan' genes are indeed orphans is very poorly done. Firstly, the authors appear to have taken as their starting point a bioinformatics analysis published in 2007 – Line 91 reads 'The *F. graminearum* genome is predicted to encode hundreds of orphan genes (ref33). Secondly, the test for orphan status appears based only from comparisons of *F. verticillioides* and *F. oxysporum*. These are certainly not the closest *Fusarium* species to *F. graminearum* (*Fg*) . *Fg* has 4 large chromosomes, whereas *Fv* has 12 and *Fo* can have up to 16. In the literature the taxonomically closest related species are *F. culmorum* and *F. venenatum* (*Fv*) . The *Fc* draft genome publically available in ENSEMBL and UNIPROT and the published fully annotated *Fv* is more widely available. Just focussing on the sequences deposited in NCBI does not represent a complete study. The bioinformatics analysis has also not take account of the fact that several other published full sequenced *Fg* genomes are available. For example, the *osp44* is absent in several strains, including the Australian strain CS3005. A study of other available *Fg* genomes for presence absence/ sequence variants is required for the *Fg* genes selece to be orphans

There is no mention of the e value cut-off for what is considered a 'homolog' in the blast searches of *F. verticillioides* and *F. oxysporum*. The authors also need to include exactly which genomes and versions they specifically used for their analyses.

Lines 435 – 437 'To our knowledge, the active adoption of competing orphan proteins in both fungal pathogen and plant hosts has not been reported in other pathosystems'. Until the bioinformatics analysis has been redone, it is not possible for the authors to draw these conclusions. Indeed the naming of this sub-set of *osp* as orphan proteins and the total number may be incorrect. Therefore the manuscript title may also need to be changed.

Generation and characterisation of the set of single gene deletion *Fg* strains in PH-1

The successful generation of fifty single gene deletions has been stated and this data is summarised in one of the Supplementary Tables. However, no molecular evidence or primer details are given that indicate to the reader that for any gene a successful deletion has been made. This molecular evidence for all 50 genes and the actual strains selected for the functional tests is required. This key information can be presented as supplementary figures

The functional testing of each Fg strain.

This reviewer is not convinced by the results for osp25 and osp44 presented in Figure 1. Although the graph of figure 1f shows a significant reduction, the photos of figure 1e does not represent the same. Also it would be good to test a 2nd wheat cultivar with the 3 lead genes to ensure that the results are re-producible between cultivars.

At this point the testing of another plant species also becomes relevant because of the results given below specifically on TaFROG. TaFROG is a Pooideae-specific orphan protein of host origin. Therefore, it would be anticipated that the lead Fg orphan effector OSP24 would not be required for virulence on any other cereal species or a non-cereal species. Infection assays on barley, maize, rice and Arabidopsis would therefore be highly informative and would be a further formal test of the main model presented from this study.

Analysis of the reduce pathogenicity phenotype of osp24 in planta – production of mycotoxin

The many factors influencing mycotoxin production is a topic of great interest to the international Fusarium community. For the reader it would be very interesting to know for the rachis tissue, for the expression of some of the TRI genes also to be included in Fig. 2c. This is because the lower biomass could mean that because the DON levels recovered did not change, then per unit fungal biomass more DON from greater TRI expression and /or less conversion to the glucosyl form by the host must be occurring. Or is this result caused because in Figure 1g the DON results are presented per diseased spikelet, whereas in Fig 2c the data is from diseased whole heads . If the latter is the case, then an additional comparable analysis on just diseased spikelets is required.

Suppression of PCD induced by BAX or INF1

In figure 2k there is a large difference in PCD induced by INF1 on the two leaves in the image suggesting an element of leaf-to-leaf variation for this assay. This variation is not referred to in the main text. A far better experimental design is to place the controls on the same leaf as the INFI + OSP24 co-expression to limit this effect.

RNA seq analysis to identify DEG in wheat heads post WT and Mutant infection.

In this section the authors focus part of their detailed analyses on 17 of the 215 up-regulated DEGs in osp24-infected wheat heads encode NBS-LRR proteins that are known to be involved in plant immunity against microbial pathogens. Can the authors explain why they included an analysis of NBS-LRR which are almost exclusively involved in gene-for-gene mediated defense activation? Whereas for the wheat-Fg interaction no race specificity has been previously reported, all resistance reported in wheat is QTL mediated and strain non-specific.

The GO analysis of the DEG is very weak. The text reads – ‘up-regulated DEGs were significantly enriched in genes functionally related to energy, metabolism, and plant defense responses’. These are all very high level categories. Describing specific GO would be far better.

In this section the DEG for Fg have not been included, even though the replicated data must be available for the RNA-Seq analyses. For completeness this data set needs to be included and analysed in case there are many other Fg effectors induced due to the lack of Fgosp24.

Y2H to identify osp24 interactors

This analysis clearly identifies the alpha sub-unit TaSnRK1α as one of the highly recovered interactor sequences. Wheat is a hexaploid species and often the A, B and D homoeologues are all expressed in specific tissues and often have high sequence homology. Therefore the authors need to explain why only the TaSnRK1α has been identified in this screen and indeed which

homoeologue has been identified.

Analysis to transgenic wheat plants overexpressing or silencing TaSnRK1 α .

This data is very interesting. But it would also be good to include the DON data associated with both sets of wheat lines and relate these results to the biomass level differences detected on a per diseased spikelet basis (see immediately below for reason)

Osp24 accelerates TaSnRK1 α degradation during infection

This dataset is generated in an interesting way, by mixing isolated proteins from WT and osp24 mutant infected wheat heads with the purified TaSNK1 α protein and exploring the in vitro rates of degradation. Is it highly likely that the levels of DON in the wheat heads have greatly influenced the range of proteins recovered from the infected wheat heads. The authors stated that 'these results suggest that Osp24 may accelerate TaSnRK1 α degradation during infection'.

(lines 279-280). But it is highly likely that the different DON effect in the wheat heads due to the lower Fg biomass is altering the composition of the total protein mix extracted which then goes into this assay. The authors need to think through what additional control experiments need to be done to test this formal possibility.

TaFROG may compete with Osp24 in binding with TaSnRK1 α and reduce its degradation via the ubiquitin-26S proteasome system

In the experiment reported in lines 346-354 only the PH-1 was considered. For completeness and to further formally test the model being proposed for osp24 function, the comparable protein samples need to be made from wheat infected with the osp24 mutant and the levels of TaSnRK1 α degradation compared to those found in the PH-1 interaction.

Discussion

Lines 363-4 . Fg does not possess supernumerary chromosomes, therefore this comment is irrelevant

Lines 412 –421 This is an additional result which had been placed in the discussion and not in the results section. If the authors wish to keep the SGT1 result in this manuscript, then this needs to be moved to the results section and correctly described. If not this observation should be fully removed from this manuscript and the necessary author adjustments made if applicable.

Line 422 onwards – Role TaFROG and/or DON in other plant species

TaFROG is DON inducible, but in the model species Arabidopsis, which is fully susceptible to Fg infections, DON is not required for virulence and the TaFROG protein is not present. Therefore what would be the alternative model in a plant species successfully infected by Fg, where neither are functionally required. The readers and this reviewer would be interested in the authors thoughts on this point.

Lines 435 – 437 To our knowledge, the active adoption of competing orphan proteins in both fungal pathogen and plant hosts has not been reported in other pathosystems.

Until the bioinformatics analysis has been redone, it is not possible for the authors to draw these conclusions. Indeed the naming of this sub-set of osp as orphan proteins and the total number may be incorrect.

Figure legends

Figure 1 Please refer to the source of the RNAseq data within the legend for Figure 1i. The current legend suggests that these transcriptomic experiments were done as part of this study which is not the case.

Methods:

- Page 23 (line 467): Include cultivar KN199 and time of inoculation.
- Page 23 (line 469): reference 80 uses number of spikelets infected. Does it mean the same as "disease index". Please clarify.

Other points

- Ln 53: change to 'effectors are able to be secreted'
- Ln 173-177: sentence needs splitting and rewording to improve readability
- Ln 178: reword to 'formation of intra-molecular disulfide bonds'
- Ln 445: give URL for 'Broad Institute website' download
- S-table 1: No gene deletion diagnostics tests given; no primer table given for creating gene deletions.
- S-table 2: no Standard deviation given
- Suppl Fig1: re. figure legend - not sure where category 'plant defense response' is in the figure. Perhaps forgotten to include. Not sure how meaningful this analysis is due to the fact that categories 'Energy, metabolism' is quite unspecific.

Reviewer #2 (Remarks to the Author):

The manuscript by Jiang et al describes the identification of proteins involved in the interaction between *Fusarium graminearum* and wheat. This is quite a breakthrough study with respect to the pathogen side of the work as there are very few small secreted proteins that have been identified to date from this pathogen that are involved in the infection, particularly when the size of the research field is considered in comparison to other pathosystems.

Not only do the authors identify 3 secreted proteins involved in virulence (and characterise one of them) they have undertaken a detailed study of the pathways in which this protein interacts with in the host (after identifying that it was translocated to the host). The authors should be commended on a fantastic piece of work that has been presented in a very succinct manner.

I have very few comments

Line 53: "...are able to be secreted..."

Line 93 and 435: I think there is some conjecture as to whether *Fusarium graminearum* and wheat actually co-evolved together (see doi 10.1111/nph.14894). Perhaps a more general description could be used such as grass host? Presumably given the data presented in this work the original host(s) of Fg might have TaFROG homologues.

Lines 411-422. Perhaps this section on TaSGT1 could be left out. It is interesting but leaving it out won't lessen the story and it could be held back for another publication.

Figure 5: I can't see a reference to it in the main body of the paper. It's a nice summary so I recommend keeping it

OSP24, 25 and 26. Perhaps on the first use of these three genes (line 129) the gene numbers could be given in brackets. Currently these are in the supplementary data but it could be useful in the main text too.

Donald Gardiner

Reviewer #3 (Remarks to the Author):

This is a very interesting and well-written study showing that an orphan secreted protein OSP24 from *Fusarium graminearum* plays a role as a virulence determinant of this very important wheat pathogen. OSP24 is specifically expressed during infection and *osp24* Fg knock-out mutants display attenuated symptom development as well as reduced growth within the host tissue. The effector is translocated into the plant host cell where it localizes to the nucleus. Upon transient expression in *N. benthamiana* OSP24 is able to suppress cell death elicited by two different cell death inducing proteins, which could point toward its function in pathogenesis. The authors screen a wheat derived yeast two-hybrid library using OSP24 as a bait to identify potential host targets. Among other proteins, the plant energy sensor kinase SnRK1 was found as binding partner for OSP24. Functional relevance of this interaction was demonstrated in transgenic wheat lines with reduced or elevated SnRK1 levels. SnRK1 RNAi lines displayed enhanced susceptibility against Fg, while overexpression lines were more tolerant. Biochemical *in vitro* data suggest that binding of OSP24 destabilizes SnRK1 by increasing its proteasomal turnover and thus interferes with SnRK1 signaling in the host. Several lines of evidence from different plants now support a role of SnRK1 in immunity and previous research has identified an orphan protein from wheat, TaFROG, as a SnRK1 binding protein. TaFROG1 expression is induced during Fg infection and transgenic overexpression enhances disease tolerance. The authors took up on these previous data and demonstrate that TaFROG and OSP24 compete to bind similar regions on the SnRK1 polypeptide. Furthermore, TaFROG antagonizes the destabilizing activity of OSP24 on SnRK1 by preventing OSP24 binding.

The strong point of the study is of course the principle finding that an orphan protein from *Fusarium* acts as a virulence factor whose function is antagonized by an orphan protein from the host. Furthermore, the identification of SnRK1 as a target protein for OSP24 represents a major step forward to our understanding about the role of this central sensor kinase in plant environment interaction. The experimental evidence to support the author's conclusion is extremely well elaborated by complementary methods and I have only a few minor remarks concerning the documentation at some points.

However, I think the weak point of the study is that it remains unclear how SnRK1 function in wheat is mechanistically linked to plant defense. Maybe this is setting the bar too high but do the authors think that the ability of OSP24 to suppress cell death is linked to its destabilizing effect on SnRK1? These two observations are currently a bit loosely linked. Another interaction partner that popped up in the Y2H was SGT1, a protein required for certain cell death responses during ETI. In the discussion the authors speculate that SGT1 could play a role in facilitating the association between SnRK1 and the degradation machinery in the presence of OSP24. However, couldn't SGT1 also be an independent target of OSP24, for instance to interfere with its cell death promoting function? Other than that, there is at least one report showing that SnRK1 is required to elicit a hypersensitive response upon recognition of a bacterial effector during ETI (Szczeny et al., *New Phytologist* (2010) 187: 1058–1074), a study that is briefly mentioned. Interestingly, this ETI response is suppressed by another effector protein that targets SnRK1. Thus, there could very well be a stronger connection between the PCD suppression by OSP24 and its ability to interfere with SnRK1 stability. Is the C-terminally truncated OSP24 which lost its ability to bind SnRK1 still able to suppress PCD? At least, I suggest the authors expand the discussion on that topic a little bit to provide some specific ideas, even speculative ones, about the connection between OSP24, PCD and SnRK1.

Minor points:

Figure 2K: Could the authors provide western blots to demonstrate BAX/INF and OSP24 coexpression?

Figure 3b: Although the quality of the figures is generally very high, the fluorescence images in the left panel of Figure 3b are just black. Even at high magnification I don't see any sign of a YFP

signal. Could these be replaced?

Page 15, line 309: "Osp24 significantly increased the amount of these proteins" Is this statement corroborated by a statistical test? If not, please avoid the term "significantly". See also page 17, line 342

Responses to Reviewers' comments:

Reviewer #1 (Remarks to the Author):

Overall this is an interesting manuscript which includes a wide range of different experiments. However, in many sections there are serious problems which the authors need to fully address.

Lineage specific adaptation

Lines 45 – 48 The authors hypothesis is that orphan genes are thought to play important roles in lineage-specific adaptation. Plant pathogenic fungi may evolve novel orphan genes to facilitate host adaptation and infection. Fungal effectors are good examples of orphan genes that have evolved for plant infection as many of them lack homologs in closely-related species. But in this manuscript there is no experimental evidence provided that the orphan gene *Osp24* provides a lineage specific response because only the host species, wheat has been tested.

Response: First of all, because orphan genes are lineage (taxonomically)-restricted genes, 'lineage-specific adaptation' described in this manuscript refers to the adaptation of the fungal lineage with the orphan genes to host plants or environments. ('Lineage-specific' relevant to different fungal lineages with or without orphan genes, not about lineages of plant hosts). Nevertheless, as suggested, we conducted infection assays with corn silks and Arabidopsis floral tissues during revision and found that the *osp24* mutant had no obvious defects in virulence in these infection assays. A figure (Supplementary Figure 4) was added to the revised manuscript to show related data in the revised manuscript. These results indicated that *Osp24* indeed facilitates the adaptation of *F. graminearum* to wheat and enhances its virulence, possibly by accelerating the degradation of TaSnRK1 α .

Bioinformatics analyses

The bioinformatics analysis done to reveal that the 50 selected 'orphan' genes are indeed orphans is very poorly done. Firstly, the authors appear to have taken as their starting point a bioinformatics analysis published in 2007 – Line 91 reads 'The *F. graminearum* genome is predicted to encode hundreds of orphan genes (ref33). Secondly, the test for orphan status appears based only from comparisons of *F. verticillioides* and *F. oxysporum*. These are certainly not the closest *Fusarium* species to *F. graminearum* (*Fg*). *Fg* has 4 large chromosomes, whereas *Fv* has 12 and *Fo* can have up to 16. In the literature the taxonomically closest related species are *F. culmorum* and *F. venenatum* (*Fv*). The *Fc* draft genome publically available in ENSEMBL and UNIPROT and the published fully annotated *Fv* is more widely available. Just focussing on the sequences deposited in NCBI does not represent a complete study. The bioinformatics analysis has also not take account of the fact that several other published full sequenced *Fg* genomes are available. For example, the *osp44* is absent in several strains, including the Australian strain CS3005. A study of other available *Fg* genomes for presence absence/ sequence variants is required for the *Fg* genes select to be orphans. There is no mention of the e value cut-off for what is considered a 'homolog' in the blast searches of *F. verticillioides* and *F. oxysporum*. The authors also need to include exactly which genomes and versions they specifically used for their analyses.

Lines 435 – 437 ‘To our knowledge, the active adoption of competing orphan proteins in both fungal pathogen and plant hosts has not been reported in other pathosystems’. Until the bioinformatics analysis has been redone, it is not possible for the authors to draw these conclusions. Indeed the naming of this sub-set of osp as orphan proteins and the total number may be incorrect. Therefore the manuscript title may also need to be changed.

Response: Sorry for the confusion. The orphan genes in our study is not obtained from the ref 33. We identified them independently using our bioinformatics analysis as described in Materials and Methods. In fact, the study reported in this manuscript was a part of the large-scale knockout project initiated in our labs back in 2010 that aimed to characterize all the genes unique to *F. graminearum*. At that time only the genome sequences of *F. verticillioides* and *F. oxysporum* were publicly available as the most closely-related species of *F. graminearum*. All the predicted protein sequences of *F. graminearum* without homologs in these two *Fusarium* species were extracted and then used as queries to search against NCBI nr database by BLASTp. The genes without detectable homologs in these searches were considered as the *F. graminearum* unique genes. We used E-value cut-off of 1e-5 in all Blast homolog searches. The genome version 3 from Broad Institute was used. We added this information in the revised manuscript.

As the reviewer pointed out, we also noted that some of these ‘unique’ genes characterized in this study have homologs in the genome sequences of other closely-related species that became available after 2010 (such as the *Fusarium* species mentioned by the reviewer). Therefore, we used the “orphan genes” instead of “unique genes” in this manuscript. Because orphans are defined as taxonomically-restricted genes and homologs of orphan genes can be present in closely related species (Khalturin et al., 2009 – this reference was cited). As previously noted (Khalturin et al., 2009), this category of genes will change in the course of further genome sequences of closely-related species become available. Orphan genes could be gained and be lost during evolution in some species or strains. For instance, the *TaFROG* gene is currently recognized as a Pooideae-specific orphan gene. However, not all species of Pooideae contain the *TaFROG* homolog.

Therefore, based on this definition, Osp24 is an orphan secretory protein in *F. graminearum*. Overall, we believe our data supported our major conclusion summarized in the title: ‘The orphan secretory protein Osp24 of *Fusarium graminearum* modulates host immunity by mediating the proteasomal degradation of TaSnRK1 α ’.

Generation and characterisation of the set of single gene deletion Fg strains in PH-1.

The successful generation of fifty single gene deletions has been stated and this data is summarised in one of the Supplementary Tables. However, no molecular evidence or primer details are given that indicate to the reader that for any gene a successful deletion has been made. This molecular evidence for all 50 genes and the actual strains selected for the functional tests is required. This key information can be presented as supplementary figures

Response: In the revised manuscript, all the primers used in this study were listed in Supplementary Table 7. PCR results to verify the gene deletion events for all 50 *OSP* genes were presented in Supplementary Figure 1.

The functional testing of each Fg strain.

This reviewer is not convinced by the results for *osp25* and *osp44* presented in Figure 1. Although the graph of figure 1f shows a significant reduction, the photos of figure 1e does not represent the same. Also it would be good to test a 2nd wheat cultivar with the 3 lead genes to ensure that the results are re-productible between cultivars.

Response: Figure 1e was revised in the revised manuscript. As suggested, we assayed the virulence of the *osp24*, *osp25*, and *osp44* mutants with wheat heads of cultivar Zhoumai 36 (Li et al., 2019), a Chinese susceptible cultivar. Same as the results from infection assays with XiaoYan 22, all these three mutants were reduced in virulence in infection assays with Zhoumai 36. In the revised manuscript, supplementary Figure 2 was added to show representative images of wheat heads of Zhoumai 36 infected with the *osp24*, *osp25*, and *osp44* mutants.

At this point the testing of another plant species also becomes relevant because of the results given below specifically on TaFROG. TaFROG is a Pooideae-specific orphan protein of host origin. Therefore, it would be anticipated that the lead Fg orphan effector OSP24 would not be required for virulence on any other cereal species or a non-cereal species. Infection assays on barley, maize, rice and Arabidopsis would therefore be highly informative and would be a further formal test of the main model presented from this study.

Response: As suggested, we conducted infection assays with corn silks and Arabidopsis floral tissues, and found that the *osp24* mutant had no obvious defects in virulence on these plants. Related results were presented in Supplementary Figure 4.

Analysis of the reduce pathogenicity phenotype of *osp24* in planta – production of mycotoxin

The many factors influencing mycotoxin production is a topic of great interest to the international Fusarium community. For the reader it would be very interesting to know for the rachis tissue, for the expression of some of the TRI genes also to be included in Fig. 2c. This is because the lower biomass could mean that because the DON levels recovered did not change, then per unit fungal biomass more DON from greater TRI expression and /or less conversion to the glucosyl form by the host must be occurring. Or is this result caused because in Figure 1g the DON results are presented per diseased spikelet, whereas in Fig 2c the data is from diseased whole heads . If the latter is the case, then an additional comparable analysis on just diseased spikelets is required.

Response: First of all, for both Fig. 1g and Fig. 2c, only the diseased spikelets at the inoculation site were sampled and assayed for DON production. (This is a common practice to compare DON production during plant infection in mutants for comparison with the wild type. Not the entire wheat head). The spikelets inoculated with the wild type and *osp24* mutant had the same disease symptoms at 14 dpi. Nevertheless, as suggested, we assayed *TRI5* expression by qRT-PCR with RNA isolated from inoculated wheat spikelets during revision. No significant differences in *TRI5* expression was observed between the wild type and *osp24* mutant (Data presented in Supplementary Figure. 3 in the revised manuscript), further indicating that *Osp24* is not directly involved into DON production during plant infection.

We are well aware that various physiological and environmental conditions affect DON biosynthesis in *F. graminearum*. However, *Osp24* is a small secretory protein that is expressed during plant infection. Besides our data showed that *Osp24* is not important for DON production

in inoculated spikelets, we also think it is impossible for such a small protein secreted by the pathogen that enters plant cells to directly affect DON production in *F. graminearum*. (Some virulence factors in *F. graminearum* may indirectly affect DON production during infection because of their effects on plant physiology or defense responses. Even so, indirect effects of these virulence factors are derived from the primary mode of action on interfering with plant physiology or defense responses. Cause vs effect)

Suppression of PCD induced by BAX or INF1

In figure 2k there is a large difference in PCD induced by INF1 on the two leaves in the image suggesting an element of leaf-to-leaf variation for this assay. This variation is not referred to in the main text. A far better experimental design is to place the controls on the same leaf as the INF1 + OSP24 co-expression to limit this effect.

Response: Infiltration assays were repeated during revision as suggested. Figure 2k was revised with new images.

RNA seq analysis to identify DEG in wheat heads post WT and Mutant infection.

this section the authors focus part of their detailed analyses on 17 of the 215 up-regulated DEGs in osp24-infected wheat heads encode NBS-LRR proteins that are known to be involved in plant immunity against microbial pathogens. Can the authors explain why they included an analysis of NBS-LRR which are almost exclusively involved in gene-for-gene mediated defense activation? Whereas for the wheat-Fg interaction no race specificity has been previously reported, all resistance reported in wheat is QTL mediated and strain non-specific.

Response: The rationale for describing these putative NBS-LRR genes in RNA-seq analysis because they were significantly enriched in the up-regulated DEGs (accounting for 7% of the up-regulated DEGs). In contrast, putative NBS-LRR genes were absent in the down-regulated DEGs. Even though there is no race-specificity in *F. graminearum*, we believe this is an interesting observation and some of these putative NBS-LRR may contribute to plant defenses against fungal infection (maybe not in a race-specific manner). Related sentences were revised to clarify about this point.

The GO analysis of the DEG is very weak. The text reads – ‘up-regulated DEGs were significantly enriched in genes functionally related to energy, metabolism, and plant defense responses’. These are all very high level categories. Describing specific GO would be far better.

Response: In the previously submitted manuscript, the FunCat functional annotation scheme was used for DEGs enrichment. In the revised manuscript, we added the GO enrichment results of the DEGs in Supplementary Table. 4. Several GO categories associated with metabolism were significantly enriched, including oxidation-reduction process, glycolytic process, glycine catabolic process, malate metabolic process, peroxisome fission, chlorophyll biosynthetic process, fructose 2,6-bisphosphate metabolic process, fructose metabolic process, L-serine metabolic process, photosynthesis and ATP synthesis coupled proton transport.

In this section the DEG for Fg have not been included, even though the replicated data must be available for the RNA-Seq analyses. For completeness this data set needs to be included and analysed in case there are many other Fg effectors induced due to the lack of Fgosp24.

Response: Fungal DEGs were analyzed as suggested. Only 54 and 83 fungal genes were up- and down-regulated in the *osp24* mutant in comparison with the wild type during wheat infection. None of putative effector genes predicted by EffectorP was induced in the *osp24* mutant. Related data were presented in Supplemental Table 5 and described in Results.

Y2H to identify osp24 interactors

This analysis clearly identifies the **alpha sub-unit** TaSnRK1 α as one of the highly recovered interactor sequences. Wheat is a hexaploid species and often the A, B and D homeologues are all expressed in specific tissues and often have high sequence homology. Therefore the authors need to explain why only the TaSnRK1 α has been identified in this screen and indeed which homoeologue has been identified.

Response: SnRK1 is a multi-subunit complex consisting of a catalytic α subunit, a regulatory β subunit and an activating γ subunit. These three subunits differed in amino acid sequences and functions. In this study, we identified TaSnRK1 α subunit as an Osp24-interacting protein in wheat. The wheat genome has three TaSnRK1 α homeologues (A, B and D) that share the same amino acid sequence but differ slightly (12 SNPs between A and B; 14 SNP between A and D). Interestingly, sequencing analysis showed that all the five TnSnRK1 α clones identified in the original yeast two-hybrid library screen were TaSnRK1 α -A. Therefore, in this study we used TnSnRK1 α -A for all the experiments related to TaSnRK1 α .

Analysis to transgenic wheat plants overexpressing or silencing TaSnRK1 α .

This data is very interesting. But it would also be good to include the DON data associated with both sets of wheat lines and relate these results to the biomass level differences detected on a per diseased spikelet basis (see immediately below for reason)

Response: During revision, we assayed DON production as suggested. In the diseased wheat spikelets inoculated with PH-1, DON production was similar (no significant difference) between KN199 and its transgenic plants. Related data were presented in Supplementary Figure 9 and described in the revised manuscript.

Osp24 accelerates TaSnRK1 α degradation during infection

This datasets is generated in an interesting way, by mixing isolated proteins from WT and *osp24* mutant infected wheat heads with the purified TaSNK1 α protein and exploring the in vitro rates of degradation. Is it highly likely that the levels of DON in the wheat heads have greatly influenced the range of proteins recovered from the infected wheat heads. The authors stated that 'these results suggest that Osp24 may accelerate TaSnRK1 α degradation during infection'. (lines 279-280). But it is highly likely that the different DON effect in the wheat heads due to the lower Fg biomass is altering the composition of the total protein mix extracted which then goes

into this assay. The authors need to think through what additional control experiments need to be done to test this formal possibility.

Response: This question is also related to DON. As explained in responses to earlier questions (see above), our data showed that Osp24 is not important for early infection processes and DON production during plant infection. Reduced virulence of the *osp24* mutant was due to its defect in infectious growth in rachis tissues. Therefore, we think this question about Osp24 may affect DON, and DON may in turn affect many plant proteins is not based on our data. In addition, the role of Osp24 on TaSnRK1 α degradation was further confirmed in degradation assays with the GST-Osp24 fusion protein and total protein extracted from healthy wheat head (Figure.4c).

TaFROG may compete with Osp24 in binding with TaSnRK1 α and reduce its degradation via the ubiquitin-26S proteasome system

In the experiment reported in lines 346-354 only the PH-1 was considered. For completeness and to further formally test the model being proposed for *osp24* function, the comparable protein samples needs to be made from wheat infected with the *osp24* mutant and the levels of TaSnRK1 α degradation compared to those found in the PH-1 interaction.

Response: As suggested, we compared the degradation of TaSnRK1a in the transgenic lines overexpressing TaFROG inoculated with PH-1 and *osp24* mutant. Resulting data were presented in Supplementary Figure 13. The rate of TaSnRK1 degradation was similar in co-incubation mixtures with proteins from *osp24*-infected or PH-1-infected wheat heads. In PH-1-infected wheat heads, a high expression level of TaFROG may enable its protection of TaSnRK1 α -A against Osp24 binding and degradation.

Discussion

Lines 363-4 . Fg does not possess supernumerary chromosomes, therefore this comment is irrelevant

Response: We are aware that the sequenced *F. graminearum* strains lack of supernumerary chromosomes. However, different field isolates of the same fungal species are known to vary in supernumerary chromosomes, and some fungal supernumerary chromosomes are unstable. Therefore, it remains possible that some field strains of *F. graminearum* out in the nature may have stable or unstable supernumerary chromosomes. We could delete this sentence but prefer to keep a revised sentence in discussion because of this possibility and the chromosomal locations of these genes.

Lines 412 –421 This is an additional result which had been placed in the discussion and not in the results section. If the authors wish to keep the SGT1 result in this manuscript, then this needs to be moved to the results section and correctly described. If not this this observation should be fully removed from this manuscript and the necessary author adjustments made if applicable.

Response: Thanks for the suggestion. We are currently conducting additional experiments to further characterize the interaction of Osp24 and TaSGT1 and determine the role/mechanism of

TaSGT1 in resistance against *F. graminearum*. Therefore, we removed this part of discussion in the revised manuscript as suggested.

Line 422 onwards – Role TaFROG and/or DON in other plant species

TaFROG is DON inducible, but in the model species *Arabidopsis*, which is fully susceptible to Fg infections, DON is not required for virulence and the TaFROG protein is not present. Therefore what would be the alternative model in a plant species successfully infected by Fg, where neither are functionally required. The readers and this reviewer would be interested in the authors thoughts on this point.

Response: This part of Discussion was revised. In *Fusarium graminearum*, DON is not essential for the initial infection. The *tri5* mutant blocked in DON biosynthesis still causes typical disease symptoms on the inoculated wheat spikelet as the wild type. However, DON is an important virulence factors for infection growth to spread via the rachis. For infection assays with *Arabidopsis*, which is not a natural host to *F. graminearum*, both floral tissues and leaves have been used. For infection assays with floral tissues (Brewer and Hammond-Kosack, 2015), fungal hyphal growth was all over floral tissues. Spreading of infection is by fungal hyphae grown on the surface of floral tissues. For infection assays with *Arabidopsis* leaves, detached leaves were inoculated with spore drops or culture blocks of *F. graminearum* and assayed for rotting of leaf tissues. In both *Arabidopsis* infection assays, there is no need for DON because there is no such as thing as spreading via rachis tissues in wheat heads. (I personally think infection assays with *Arabidopsis* are questionable, particularly infection assays with detached leaves. Like the reported infection of *Arabidopsis* leaves by *Magnaporthe oryzae*, dead plant tissues may be degraded by saprophytic growth of fungal hyphae – not necessarily infectious growth).

Lines 435 – 437 To our knowledge, the active adoption of competing orphan proteins in both fungal pathogen and plant hosts has not been reported in other pathosystems.

Until the bioinformatics analysis has been redone, it is not possible for the authors to draw these conclusions. Indeed the naming of this sub-set of osp as orphan proteins and the total number may be incorrect.

Response: This sentence was revised and expanded.

Figure legends

Figure 1 Please refer to the source of the RNAseq data within the legend for Figure 1i. The current legend suggests that these transcriptomic experiments were done as part of this study which is not the case.

Response: Figure 1 legend was revised as suggested. A reference described the transcriptomic experiments was added.

Methods:

- Page 23 (line 467): Include cultivar KN199 and time of inoculation.

Response: Revised as suggested.

- Page 23 (line 469): reference 80 uses number of spikelets infected. Does it mean the same as “disease index”. Please clarify.

Response: Revised as suggested.

Other points

Ln 53: change to 'effectors are able to be secreted'

Response: Revised as suggested.

Ln 173-177: sentence needs splitting and rewording to improve readability

Response: This sentence was revised.

Ln 178: reword to 'formation of intra-molecular disulfide bonds'

Response: Revised as suggested.

Ln 445: give URL for 'Broad Institute website' download

Response: This project was initiated before the Broad Institute site was taken down. The Broad Institute has deposited the *F. graminearum* genome data at: <ftp://ftp.broadinstitute.org/pub/annotation/fungi/fusarium/>. This information was added in the revised manuscript.

S-table 1: No gene deletion diagnostics tests given; no primer table given for creating gene deletions.

Response: A supplemental table (Supplemental Table 7) was added to list all the primers used in this study.

S-table 2: no Standard deviation given

Response: Revised to add standard deviation.

Suppl Fig1: re. figure legend - not sure where category 'plant defense response' is in the figure. Perhaps forgotten to include. Not sure how meaningful this analysis is due to the fact that categories 'Energy, metabolism' is quite unspecific.

Response: This figure legend was revised as suggested. We also did a more specific GO analysis as suggested. Related data were presented in Supplementary Table 4.

Reviewer #2 (Remarks to the Author):

The manuscript by Jiang et al describes the identification of proteins involved in the interaction between *Fusarium graminearum* and wheat. This is quite a breakthrough study with respect to the pathogen side of the work as there are very few small secreted proteins that have been identified to date from this pathogen that are involved in the infection, particularly when the size of the research field is considered in comparison to other pathosystems.

Not only do the authors identify 3 secreted proteins involved in virulence (and characterise one of them) they have undertaken a detailed study of the pathways in which this protein interacts with in the host (after identifying that it was translocated to the host). The authors should be commended on a fantastic piece of work that has been presented in a very succinct manner.

I have very few comments

Response: Thanks.

Line 53: "...are able to be secreted..."

Response: Revised as suggested.

Line 93 and 435: I think there is some conjecture as to whether *Fusarium graminearum* and wheat actually co-evolved together (see doi 10.1111/nph.14894). Perhaps a more general description could be used such as grass host? Presumably given the data presented in this work the original host(s) of Fg might have TaFROG homologues.

Response: Revised as suggested. "Wheat" was changed into "host" in the revised manuscript.

Lines 411-422. Perhaps this section on TaSGT1 could be left out. It is interesting but leaving it out won't lessen the story and it could be held back for another publication.

Response: Thanks for the suggestion. We are currently conducting additional experiments to further characterize the interaction of Osp24 and TaSGT1 and determine the role/mechanism of TaSGT1 in resistance against *F. graminearum*. Therefore, we removed this part of discussion in the revised manuscript as suggested.

Figure 5: I can't see a reference to it in the main body of the paper. It's a nice summary so I recommend keeping it

Response: Figure 5 was cited in the revised manuscript.

OSP24, 25 and 26. Perhaps on the first use of these three genes (line 129) the gene numbers could be given in brackets. Currently these are in the supplementary data but it could be useful in the main text too.

Response: Revised as suggested. Gene numbers were added.

Reviewer #3 (Remarks to the Author):

This is a very interesting and well-written study showing that an orphan secreted protein OSP24 from *Fusarium graminearum* plays a role as a virulence determinant of this very important wheat pathogen. OSP24 is specifically expressed during infection and *osp24* Fg knock-out mutants display attenuated symptom development as well as reduced growth within the host tissue. The effector is translocated into the plant host cell where it localizes to the nucleus. Upon transient expression in *N. benthamiana* OSP24 is able to suppress cell death elicited by two different cell death inducing proteins, which could point toward its function in pathogenesis. The authors screen a wheat derived yeast two-hybrid library using OSP24 as a bait to identify potential host targets. Among other proteins, the plant energy sensor kinase SnRK1 was found as binding partner for OSP24. Functional relevance of this interaction was demonstrated in transgenic wheat lines with reduced or elevated SnRK1 levels.

SnRK1 RNAi lines displayed enhanced susceptibility against Fg, while overexpression lines were more tolerant. Biochemical *in vitro* data suggest that binding of OSP24 destabilizes SnRK1 by increasing its proteasomal turnover and thus interferes with SnRK1 signaling in the host. Several lines of evidence from different plants now support a role of SnRK1 in immunity and previous research has identified an orphan protein from wheat, TaFROG, as a SnRK1 binding protein. TaFROG1 expression is induced during Fg infection and transgenic overexpression enhances disease tolerance. The authors took up on these previous data and demonstrate that TaFROG and OSP24 compete to bind similar regions on the SnRK1 polypeptide. Furthermore, TaFROG antagonizes the destabilizing activity of OSP24 on SnRK1 by preventing OSP24 binding.

The strong point of the study is of course the principle finding that an orphan protein from *Fusarium* acts as virulence factor whose function is antagonized by an orphan protein from the host. Furthermore, the identification of SnRK1 as a target protein for OSP24 represents a major step forward to our understanding about the role of this central sensor kinase in plant environment interaction. The experimental evidence to support the author's conclusion is extremely well elaborated by complementary methods and I have only a few minor remarks concerning the documentation at some points.

However, I think the weak point of the study is that it remains unclear how SnRK1 function in wheat is mechanistically linked to plant defense. Maybe this is setting the bar too high but do the authors think that the ability of OSP24 to suppress cell death is linked to its destabilizing effect on SnRK1? These two observations are currently a bit loosely linked. Another interaction partner that popped up in the Y2H was SGT1, a protein required for certain cell death responses during ETI. In the discussion the authors speculate that SGT1 could play a role in facilitating the association between SnRK1 and the degradation machinery in the presence of OSP24. However, couldn't SGT1 also be an independent target of OSP24, for instance to interfere with its cell death promoting function? Other than that, there is at least one report showing that SnRK1 is required to elicit a hypersensitive response upon recognition of a bacterial effector during ETI (Szczeny et al., *New Phytologist* (2010) 187: 1058–1074), a study that is briefly mentioned. Interestingly, this ETI response is suppressed by another effector protein that targets SnRK1. Thus, there could very well be a stronger connection between the PCD suppression by OSP24 and its ability to interfere with SnRK1 stability. Is the C-terminally

truncated OSP24 which lost its ability to bind SnRK1 still able to suppress PCD? At least, I suggest the authors expand the discussion on that topic a little bit to provide some specific ideas, even speculative ones, about the connection between OSP24, PCD and SnRK1.

Response: Thanks for the comments and suggestions. Regarding the comment about the link with SnRK1, we revised the discussion to be more comprehensive.

Regarding the comment on TaSGT1, we are currently conducting additional experiments to further characterize the interaction of Osp24 and TaSGT1 and determine the role/mechanism of TaSGT1 in resistance against *F. graminearum*. Therefore, as suggested by reviewer #1 and reviewer #2, we removed this part of discussion in the revised manuscript. We also revised discussions related to the connection between OSP24, PCD and SnRK1, and added reference as suggested.

Regarding the question about the C-terminal truncation of Osp24, as suggested, we assayed the effect of C-terminal truncation of Osp24 on PCD suppression. Truncation of the C-terminal region of Osp24 blocked its suppression of Bax- or INF1-induced PCD, indicating that the ability to bind with TaSnRK1 α is essential for Osp24 to suppress PCD. Data were presented in Supplementary Figure 7 in the revised manuscript.

Minor points:

Figure 2K: Could the authors provide western blots to demonstrate BAX/INF and OSP24 coexpression?

Response: Revised as suggested. Results from western blot analysis were added in Supplementary Figure 5.

Figure 3b: Although the quality of the figures is generally very high, the fluorescence images in the left panel of Figure 3b are just black. Even at high magnification I don't see any sign of a YFP signal. Could these be replaced?

Response: This comment is related to the negative controls. With only TaSnRK1 α -nYFP or Osp24-cYFP was expressed, no YFP signals was observed in these negative controls. The figure legend was revised to present better explanations about the negative control.

Page 15, line 309: "Osp24 significantly increased the amount of these proteins" Is this statement corroborated by a statistical test? If not, please avoid the term "significantly". See also page 17, line 342.

Response: The word "significantly" was removed.

REVIEWERS' COMMENTS:

Reviewer #1 (Remarks to the Author):

The authors have done very well to answer all the questions and concerns raised about their initial manuscript. They have also added the requested missing information and experimental data.

Upon rereading the revised manuscript, just a few points were noted which are given below.

Missing info, problem with figure panel and missing scripts

L197 and again L311 – add the days post *Fusarium* inoculation the total protein extracts were made from the wheat heads for these assays.

Figure 4 Panel H, KN177 compared to Ta FROG OE wheat line. Why is there also FHB disease symptoms at the top of the inoculated head, when this was a point inoculation exp? In the method section it states - wheat heads were with 10 µl of conidium suspensions at the 5th spikelet from the base. This image should be replaced.

New Ln. 552::

“Functional Catalogue (FunCat) and Gene Ontology (GO) enrichment annotations were performed with custom scripts and Blast2GO , respectively.” => Authors should provide custom scripts in supplemental or deposit to <https://zenodo.org/> to obtain citable DOI.

Suggested changes to the text

L117 change to ` by using the bioinformatics analyses described in the Methods section`

L142 the Ref Peng et al 2019 should be presented as a numbered reference

L276 change ` responsible for` to ` involved in`

L277 change to ` expression of Ops24 with a truncated C-terminal region...`

L306 change ` but` to `and`

L485 change to ` to enhance FHB resistance independent...`

L488 change to ` and each may ...`

L511 -L512 , change to ` that late became available in the public domain.`

L522 change ` they` to `these sequences`

L523 change to ` these predicted orphan proteins...`

L533 change to at `7 or 14 days` or at `14 or 17 days` which ever is correct

Discussion

Does this not contain anything about a possible uptake mechanism in either N. benthamiana or wheat ? - re results lines – L211 -223 . This point should be briefly discussed, because the target for this new Fg effector is definitely intracellular located.

Reviewer #2 (Remarks to the Author):

All my previous comments have been suitably addressed. I have no further suggestions.
Donald Gardiner

Reviewer #3 (Remarks to the Author):

The authors have done an excellent job in revising the manuscript according to the remarks raised by the reviewers. The additional data now included in the study further support the conclusions drawn.

I have a few very minor points:

Line 141, "Supplementary Figure 2" should likely be "Supplementary Figure 3"

Line 278/279 "These results indicate that the C-terminal region of Osp24 and its interaction with SnRK1 may be essential for its function to suppress PCD." This might be too strong a statement. The data show that the region of Osp24 responsible for PCD suppression and interaction with SnRK1 overlap. This opens the possibility that SnRK1 binding could be related to its function in PCD suppression; however, both functions could still be independent. Maybe the statement should be toned down accordingly (see also line 467 in discussion)

line 485 "FHB" should probably be "FHB resistance"

Point-by-point response to reviewers' comments:

Reviewer #1 (Remarks to the Author):

The authors have done very well to answer all the questions and concerns raised about their initial manuscript. They have also added the requested missing information and experimental data. Upon rereading the revised manuscript, just a few points were noted which are given below. Missing info, problem with figure panel and missing scripts

Response: Thanks. We appreciate your suggestions to improve this manuscript.

L197 and again L311 – add the days post Fusarium inoculation the total protein extracts were made from the wheat heads for these assays.

Response: Added as suggested.

Figure 4 Panel H, KN177 compared to Ta FROG OE wheat line. Why is there also FHB disease symptoms at the top of the inoculated head, when this was a point inoculation exp? In the method section it states - wheat heads were with 10 µl of conidium suspensions at the 5th spikelet from the base. This image should be replaced.

Response: Figure 4h was revised by replacing the questionable image with a new image in the revised manuscript.

New Ln. 552::

“Functional Catalogue (FunCat) and Gene Ontology (GO) enrichment annotations were performed with custom scripts and Blast2GO, respectively.” => Authors should provide custom scripts in supplemental or deposit to <https://zenodo.org/> to obtain citable DOI.

Response: The custom script has been deposited at github.com. The link <https://github.com/xulab-nwafu/funcat> was added in this sentence and code availability in the revised manuscript.

Suggested changes to the text

L117 change to ‘ by using the bioinformatics analyses described in the Methods section’

Response: Revised as suggested.

L142 the Ref Peng et al 2019 should be presented as a numbered reference

Response: Revised as suggested.

L276 change ‘ responsible for’ to ‘ involved in’

Response: Revised as suggested.

L277 change to ‘ expression of Ops24 with a truncated C-terminal region...’

Response: Revised as suggested.

L306 change ' but' to 'and'

Response: Revised as suggested.

L485 change to ' to enhance FHB resistance independent....'

Response: Revised as suggested.

L488 change to ' and each may ...'

Response: Revised as suggested.

L511 -L512 , change to ' that late became available in the public domain.'

Response: Revised as suggested.

L522 change ' they' to 'these sequences'

Response: Revised as suggested.

L523 change to ' these predicted orphan proteins...'

Response: Revised as suggested.

L533 change to at '7 or 14 days' or at '14 or 17 days' which ever is correct

Response: Revised to '7 or 14 days'.

Discussion

Does this not contain anything about a possible uptake mechanism in either N. benthamiana or wheat ? - re results lines – L211 -223 . This point should be briefly discussed, because the target for this new Fg effector is definitely intracellular located.

Response: Discussion was expanded as suggested.

Reviewer #2 (Remarks to the Author):

All my previous comments have been suitably addressed. I have no further suggestions.

Donald Gardiner

Response: Thanks.

Reviewer #3 (Remarks to the Author):

The authors have done an excellent job in revising the manuscript according to the remarks raised by the reviewers. The additional data now included in the study further support the conclusions drawn.

I have a few very minor points:

Line 141, "Supplementary Figure 2" should likely be "Supplementary Figure 3"

Response: Corrected. Thanks.

Line 278/279 "These results indicate that the C-terminal region of Osp24 and its interaction with SnRK1 may be essential for its function to suppress PCD."

This might be too strong a statement. The data show that the region of Osp24 responsible for PCD suppression and interaction with SnRK1 overlap. This opens the possibility that SnRK1 binding could be related to its function in PCD suppression; however, both functions could still be independent. Maybe the statement should be

toned down accordingly (see also line 467 in discussion)

Response: Revised as suggested.

line 485 "FHB" should probably be "FHB resistance"

Response: Revised as suggested.